# Development of chimeric peptides to facilitate the neutralisation of lipopolysaccharides during bactericidal targeting of multidrug-resistant *Escherichia coli*

Zhenlong Wang[1,2,4], Xuehui Liu[3,4], Da Teng[1,2], Ruoyu Mao[1,2], Ya Hao[1,2], Na Yang[1,2], Xiao Wang[1,2], Zhanzhan Li[1,2], Xiumin Wang[1,2]* & Jianhua Wang[1,2]*

Pathogenic *Escherichia coli* can cause fatal diarrheal diseases in both animals and humans. However, no antibiotics or antimicrobial peptides (AMPs) can adequately kill resistant bacteria and clear bacterial endotoxin, lipopolysaccharide (LPS) which leads to inflammation and sepsis. Here, the LPS-targeted smart chimeric peptides (SCPs)-A6 and G6 are generated by connecting LPS-targeting peptide-LBP14 and killing domain-N6 via different linkers. Rigid and flexible linkers retain the independent biological activities from each component. SCPs-A6 and G6 exert low toxicity and no bacterial resistance, and they more rapidly kill multiple-drug-resistant *E. coli* and more effectively neutralize LPS toxicity than N6 alone. The SCPs can enhance mouse survival more effectively than N6 or polymyxin B and alleviate lung injuries by blocking mitogen-activated protein kinase and nuclear factor kappa-B p65 activation. These findings uniquely show that SCPs-A6 and G6 may be promising dual-function candidates as improved antibacterial and anti-endotoxin agents to treat bacterial infection and sepsis.

[1] Gene Engineering Laboratory, Feed Research Institute, Chinese Academy of Agricultural Sciences, 100081 Beijing, People's Republic of China. [2] Key Laboratory of Feed Biotechnology, Ministry of Agriculture and Rural Affairs, 100081 Beijing, People's Republic of China. [3] Institute of Biophysics, Chinese Academy of Sciences, Beijing, China. [4] These authors contributed equally: Zhenlong Wang, Xuehui Liu. *email: wangxiumin@caas.cn; wangjianhua@caas.cn

Gram-negative pathogenic *Escherichia coli* can cause outbreaks of diarrheal diseases in both animals and humans[1]. Globally, ~1.7 billion cases of diarrheal disease occur, killing 760,000 children every year[2] and US$ 6.9 billion in losses for farmers and industries. *E. coli* that cannot be killed by the last resort antibiotic–colistin has been found in samples from animals, meat products and patients in China[3]. Lipopolysaccharides (LPSs), also termed endotoxins, are a major component of the outer membranes of Gram-negative bacteria and are released from the cell wall during bacterial growth[4]. LPS plays a key role in the pathophysiology of sepsis and shock[5,6]. Simultaneously, LPS is also a prime natural barrier that can protect bacteria from attack by drugs[5,7]. Although antibiotics have a rapid antibacterial effect, they have some shortcomings, including the development of bacterial resistance, weak LPS-neutralizing capacity and stimulating a 3–20-fold acceleration in the release of LPS into the bloodstream, which can induce various pro-inflammatory responses[8,9]. To date, no antibiotics can adequately treat sepsis[10]. Therefore, it is very necessary to find novel candidates that can clean the battlefield after killing the bacteria, including neutralizing the LPS toxicity and antagonizing the downstream cascade.

Recently, increasing attention has been given to antimicrobial peptides (AMPs) due to their broad-spectrum antimicrobial activity and low level of induced bacterial resistance[11,12]. However, these broad-spectrum AMPs may disrupt the normal flora of the body and can lead to numerous adverse side effects[13]. Therefore, the activities against the desired bacterium of some AMPs have been specifically improved by attaching a targeting region to generate novel, specifically targeted chimeric peptides (CPs) with little impact on the normal flora; these can contain functionally independent targeting and killing domains[13,14]. It has been demonstrated that some CPs such as G10KHc, M8(KH)-20, M8-33, S6L3-33, and Syn-GNU7 can enhance selectivity and improve in vitro killing activity against targeted bacteria[13–16]. However, these studies only provide a basis for the technology in which target-specific CPs were generated against some limited bacterial species, and little attention has been given to their toxicity, resistance, in vivo antibacterial/anti-endotoxic activity.

The successful construction of CPs requires indispensable functional elements and linkers that play a vital role in improving the folding, stability and intrinsic biological activities[17]. Empirical linkers are generally classified into in vivo cleavable, flexible, and rigid linkers. Cleavable linkers, cleaved by proteases under certain physiological conditions, are commonly applied in fusion proteins to target tumor sites[17–19]. Flexible linkers (($GS)_n$ or $(G)_n$) are most commonly used in CPs such as Syn-GNU7 and LHP7 to increase the spatial separation between two domains due to their flexibility[16,20]. Comparably, rigid linkers (($EA_3K)_n$ or $(XP)_n$) have also been successfully applied to construct fusion proteins, to retain a fixed distance between the functional domains, which may be more efficient than the flexible linkers[21,22]. However, to our knowledge, thus far, no study has been reported for the rigid linkers used in AMPs.

The LBP14 peptide (residues 86–99 of a serum glycoprotein, lipopolysaccharide binding protein (LBP)) can retain significant binding ability to LPS and inhibit the binding of LPS to LBP[23,24]. Moreover, a marine AMP-N6 displays potent bactericidal activity and can neutralize LPS[25]. Meanwhile, bacterial resistance is not developed against N6, but it exhibits some cytotoxicity[25]. Here, the smart CPs (SCPs)-A6 (pdb ID: 6K4W) and G6 (pdb ID: 6K4V) are generated by connecting LBP14 (targeting domain) with N6 (killing domain) via a rigid linker (($EA_3K)_2$) and a flexible linker ($G_4S$), respectively (Fig. 1; Supplementary Figs. 1 and 2). The SCPs have low toxicity toward eukaryocytes and no bacterial resistance within 30 d. The SCPs displayed more potent antibacterial activity against multidrug resistance (MDR) *E. coli*

CVCC195 and anti-endotoxin in vitro and in vivo, indicating that both linkers retain the independent biological activities from each domain.

## Results

### Design, physicochemical properties, and structures of SCPs.
The SCPs-A6 and G6 were designed by connecting N6 with LBP14 via either a rigid linker ($(EA_3K)_2$) or a flexible linker $G_4S$ (Fig. 1). LBPN6, A6, and G6 (+9) have more positive charges, higher α-helix content and larger positively charged cloud than parents (Supplementary Table 1 and Supplementary Fig. 3a), indicating their more potent interaction with bacterial membranes;[26,27] but their hydrophobicity, aliphatic index and Boman index were lower than that of N6. Additionally, the scramble peptides had similar properties, with the exception of α-helix content (Supplementary Table 1 and Supplementary Fig. 3).

The nuclear magnetic resonance (NMR) analysis showed that the $(EA_3K)_2$ and $G_4S$ linker retained a helical conformation in A6 (ranging from E15-K24) and random coil in G6 (G15-S19) (Supplementary Fig. 3b, 4, 5 and Table 1). The structures of the killing domains in A6 and G6 were almost identical to that of the free N6 according to the similarity of the nuclear Overhauser effects (NOEs) in this region. The relative orientation between the killing and targeting domains of A6 and G6 was randomly distributed in space. Additionally, structures of scramble peptide controls, N6CK, A6CK and G6CK, were similar to those of N6, A6, and G6, respectively (Supplementary Fig. 3c). The nuclear Overhauser effect spectroscopy (NOESY) spectra and DANGLE analysis of A6 and G6 indicated that while the killing domain retained β-sheet structures, the LPS-targeting LBP14 domain displayed a random coil conformation without structural convergence in aqueous solutions (Supplementary Figs. 4 and 5).

### Activities, stability, resistance, and toxicity of the SCPs.
The minimum inhibitory concentrations (MICs) of A6 and G6 (0.88–3.52 μM) did not show improvement when compared to N6 (0.4–1.6 μM) against *E. coli* and *Salmonella*, but were lower than the MICs of LBP14 (4.53–18.1 μM) (Table 2). The MICs of A6 and G6 against *S. aureus*, *Enterococcus faecalis*, and *Streptococcus suis* were greater than 3.52 μM, which were lower than those of N6, indicating higher activity than N6. The MICs of A6, G6, and N6 against *Bacillus subtilis* were 1.5, 1.76, and 0.4 μM, respectively, indicating their decreased toxicity toward probiotics. However, A6, G6, and N6 did not significantly inhibit *Listeria ivanovii* or *Candida albicans*. Comparably, parent and scramble peptides displayed moderate, weak or no activity (Table 2). These findings suggest that the antibacterial activity of the SCPs is associated with the killing domain (N6) and the linkers.

The bactericidal kinetics showed that similar to polymyxin B (PMB), a 1× or 2× MIC of A6 and G6 could completely kill *E. coli* CVCC195 within 1 h, which is more rapid than N6 (4 h) (Fig. 2a), indicating their rapid bactericidal effect against *E. coli*.

The selectivity killing result showed that both A6 and G6 selectively killed *E. coli* in the mixed *E. coli*-*S. aureus* cultures. G6 displayed higher activity than A6 (Supplementary Fig. 6). After treatment for 1 or 5 min with A6 or G6, the ratio of *E. coli* to *S. aureus* cells was 0.77–0.59 and 0.51–0, significantly lower than that of the N6-treated group (0.91–0.67). This indicates that SCPs-A6 and G6 can preferentially kill the targeted bacteria in the mixed species cultures due to the attachment of the LPS-targeting domain LBP14.

After treatment at different temperatures (20–100 °C) for 1 h and at different pH values (2–10) for 3 h, both A6 and G6 retained their intrinsic antibacterial activity against MDR *E. coli* CVCC195 (Fig. 2b, c), indicating their stability to high temperatures and acidic/alkaline conditions. Similar to N6,

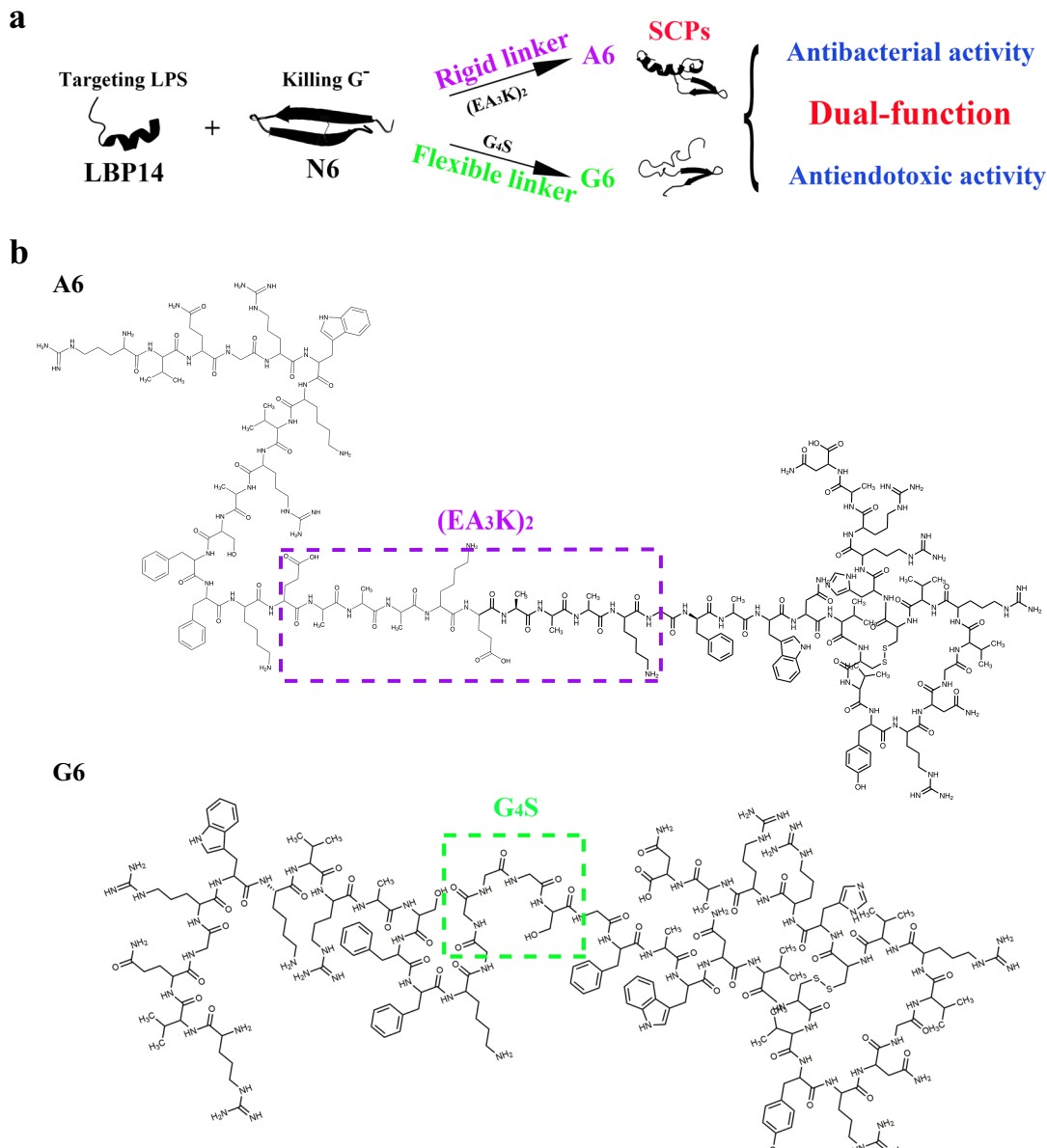

**Fig. 1 Design and structure scheme of SCPs-A6 and G6. a** Design schemes of SCPs by connecting LBP14 (a targeting domain) and N6 (a killing domain) via (EA$_3$K)$_2$/G$_4$S linkers. **b** Structure schemes of A6 and G6.

SCPs-A6 and G6 were resistant to pepsin but sensitive to trypsin (Fig. 2d). The antibacterial activities of A6, G6, and N6 slightly decreased in serum (Fig. 2e).

After a 1–2 h incubation with serum, only 27–7% and 68.5–29.5% of A6 and G6, respectively, remained intact in native patterns, which was lower than for N6 (93–87.5%) (Fig. 2f). Both A6 and G6 displayed half-lives of <2 h and almost fully degraded after 4 h (2.8% and 5.1% remaining, respectively); N6 was notably more resistant to serum (66%). In addition, over the course of 30 serial passages, the MICs of A6, G6, and PMB hardly changed, but resistance to gentamicin rapidly developed at 7 d and 15 d, with a MIC elevation of 3-fold and 15-fold (Fig. 2g). SCPs-A6 and G6 also exhibited very low hemolysis even at concentrations up to 256 µg per ml (1.2% and 0.026%, respectively), lower than N6 alone (1.9%) (Fig. 2h). The cell survival of RAW 264.7 cells when exposed to A6, G6, and N6 was 100%, 81.1%, and 88.2%, respectively, at a concentration of 128 µg per ml (Fig. 2i)[25], indicating that A6 had markedly less cytotoxicity than G6 and N6.

**SCPs induce morphologic changes in *E. coli*.** The scanning electron microscopy (SEM) images revealed that a normal cell morphology and intact cell surface were observed in the untreated *E. coli* CVCC195 cells (Fig. 3a). However, after treatment with 4 × MIC SCPs-A6 and G6 for 2 h, filamentous and extracellular polymeric substances appeared outside the cells, along with the leakage of contents. Noticeably, some collapsed lamellar cells were seen upon N6 treatment. In the PMB-treated group, numerous protrusions or blebs and filiferous substances appeared outside of the cells (Fig. 3a). For MDR *S. aureus* ATCC43300, after treatment with A6, G6, or N6, many blebs, content leakage, and lamellar collapses were observed outside of cells (Supplementary Fig. 7); in the untreated and PMB-treated controls, normal cell morphology and intact cell surfaces were apparent. This implied that the mode of action of SCPs-A6 and G6 against MDR *E. coli* and *S. aureus* is different from that of N6 or PMB.

In the control group, the *E. coli* CVCC195 cells remained a normal cellular shape and uniform cytoplasmic electron density in transmission electron microscopy (TEM) (Fig. 3b).

**Table 1 NMR and refinement statistics for SCPs-A6 and G6 structures.**

| | A6 | G6 |
|---|---|---|
| **NMR distance and dihedral constraints** | | |
| Distance constraints | | |
| Total NOE | 392 | 336 |
| Unambiguous | 317 | 274 |
| Intra-residue | 170 | 169 |
| Inter-residue | 147 | 105 |
| Sequential ($|i - j| = 1$) | 88 | 75 |
| Medium-range ($1 < |i - j| < 5$) | 13 | 10 |
| Long-range ($|i - j| >= 5$) | 46 | 20 |
| Ambiguous | 75 | 62 |
| Total dihedral angle restraints | | |
| φ | 37 | 25 |
| ψ | 37 | 25 |
| **Structure statistics** | | |
| Violations (mean and s.d.) | | |
| Distance constraints (Å) | 0.0317 ± 0.00339 | 0.0163 ± 0.00270 |
| Dihedral angle constraints (º) | 0.524 ± 0.174 | 0.311 ± 0.163 |
| Deviations from idealized geometry | | |
| Bond lengths (Å) | 0.00341 ± 0.000202 | 0.00317 ± 0.000131 |
| Bond angles (º) | 0.450 ± 0.0194 | 0.407 ± 0.0251 |
| Impropers (º) | 1.07 ± 0.173 | 0.852 ± 0.134 |
| Average pairwise r.m.s. deviation[a] (Å) | | |
| Heavy | 1.96[b] | 1.95[c] |
| Backbone | 0.61[b] | 0.68[c] |

[a]Calculated using the ordered residues in the killing domain of A6 (residue 29–43) and G6 (residue 24–38).
[b]Pairwise r.m.s. deviation was calculated among 20 refined structures.
[c]Pairwise r.m.s. deviation was calculated among 10 refined structures.

After treatment with $4 \times$ MIC of peptides for 2 h, heterogeneous electron density, disappearance of the outer and inner cell membranes, leakage of cellular contents and ghosts were observed in *E. coli*. Abnormal cells were observed with A6 and G6 treatment, which were higher than with N6 treatment, indicating their interaction with bacterial membranes, the subsequent leakage of cell contents and lysis[28]. However, the least amount of cell lysis and ghosts was seen in response to PMB treatment. The results indicated that there may be differences in how peptides and PMB interact with the *E. coli* cell membrane[28,29].

**SCPs interact with LPS**. LPS can form large micellar aggregates in water, and fluorescein isothiocyanate (FITC) fluorescence is quenched in FITC-labeled LPS micelles; peptide binding to LPS leads to the dissociation of LPS aggregates and an increase in fluorescence[5,30]. As shown in Fig. 4a, SCPs-A6 and G6 induced a larger fluorescence change than N6 and PMB in a dose-dependent manner, and G6 dissociated LPS aggregates with a greater efficacy than A6. The interaction of the SCPs with LPS may result in far more dissociation of the LPS aggregates than interaction with N6 or PMB, making it unavailable for LBP binding[5,31].

The peptide secondary structures in 10 mM Tris buffer were predominantly characterized by antiparallel strands (35.45–46.94%), α-helices (10.97–13.46%), and β-turns (15.09–17.18%), with a characteristic positive peak at 225 nm and a negative peak at 200 nm (Fig. 4b; Supplementary Table 2). In the presence of LPS, the spectra of G6 and N6 displayed two positive peaks at 195/230 nm and one negative peak at 215 nm, indicating that β-turn structures may occur in their LPS-bound states to fit each other[32,33]. However, A6 had no positive peak at 195 nm. After binding to LPS, the antiparallel structure of A6 and G6 was decreased by 12.23% and 11.46%, lower than N6 (45.21%). In contrast, random coils of A6, G6, and N6 were increased by ~7%, 6.97%, and 26.7%, and their helices were elevated by 3.76%, 3.06%, and 19.84%, respectively (Fig. 4b; Supplementary Table 2). The pronounced conformation transition of A6 and G6 from sheets to helical structures, turns and random coils occurred in the peptide-LPS complex, suggesting a strong interaction between the SCPs and LPS[34]. Either less or no change was observed in the scramble

peptides compared to that of the SCPs upon binding to LPS (Supplementary Table 2 and Supplementary Fig. 8).

The possible interaction between peptides and LPS was assessed by molecular docking (MD). Supplementary Fig. 9 shows strong networks of hydrogen bonds between the phosphates of LPS and the positively charged residues of A6 and G6. Six hydrogen bonds were formed between A6 and LPS at Lys19–PA11000:O5, Lys19–DPO2000:O1, Trp28–DPO2000:O7, Asn29–DPO2000:O1, Arg38–DPO2004:O6,O7, and Arg38–GMH1004:O7; these were stronger than those of N6[25]. Nine hydrogen bonds were formed between G6 and LPS at Arg1–DPO2000:O5, Arg1 (HH12)–DPO2000:O2, Arg1 (HH21)–DPO2000:O2, Lys7–GMH1004:O7, Lys14–DPO2004:O6, Lys14–DPO2004:O5, Arg37–PO42004:O2, Arg37–GMH1005:O6, and Arg37–PO42004:O4. Comparably, fewer (two or three) hydrogen bonds were formed between LPS and LBP14 (Lys7 (HZ1)–GMH1005:O6 and Arg9 (HH21)–MYR1014:O2), LBPN6 (Lys14 (HZ1)–FTT1013:O2, Ala34 (HN)–KDO1003:O8 and Asn35 (HD22)–KDO1003:O5), A6CK (Arg5 (HH11)–PA11000:O4 and Arg5 (HH22)–KDO1003:O1A), and G6CK (Arg9 (HH21)–GLC1007:O6 and Arg9 (HH22)–GLC1007:O4) (Supplementary Fig. 10). More LPS-binding sites of A6 (Lys19/Trp28/Asn29/Arg38) and G6 (Arg1/Lys7/Lys14/Arg37) were predicated to bind to LPS than of N6 (Arg10/Arg19/Asn21), LBP14 (Lys7/Arg9), LBPN6 (Lys14/Ala34), A6CK (Arg5), or G6CK (Arg9) (Supplementary Figs. 9 and 10), indicating the increased affinity of A6 and G6 to LPS compared to the control peptides[25].

The BODIPY-TR-cadaverine (BC) displacement assay was used to further determine the ability of peptides to interact with LPS[25]. As shown in Fig. 4c, ampicillin did not displace BC probe from its binding to LPS. SCPs-A6 and G6 induced a concentration-dependent BC displacement. The displacement ratios of A6 and G6 were both 100% at a concentration of 100 μM, which were higher than those of N6 (90.5%), PMB (83.7%), and LBP14 (74.9%), indicating a stronger binding capacity of the SCPs than N6 or LBP14, which may be ascribed to the significant binding of LBP14 to LPS[24].

To further confirm the affinity of peptides to LPS or lipid A, surface plasmon resonance (SPR) was used to monitor their real-time interactions. All peptides except N6CK could bind to LPS or lipid A in a concentration-dependent manner (Supplementary Figs. 11 and 12). The ka, kd, and KD values of A6 binding to LPS were $1.56 \times 10^3$ 1 per Ms, $1.20 \times 10^{-3}$ 1 per s, and $7.68 \times 10^{-7}$ M, respectively, which were superior to those of G6 ($1.47 \times 10^3$ 1 per Ms, $1.24 \times 10^{-3}$ 1 per s, $8.46 \times 10^{-7}$ M) and N6 ($1.03 \times 10^3$ 1 per Ms, $4.86 \times 10^{-3}$ 1 per s, $4.71 \times 10^{-6}$ M) (Supplementary Table 3), indicating that both A6 and G6 remain associated with LPS by a 20-fold stronger attraction compared to N6. In addition, the ability of A6 and G6 to bind lipid A was greater than that of LPS, suggesting that the SCPs have strong binding interactions with the lipid A domain of LPS.

In the NOESY spectra, enhanced signal intensity was observed in the presence of LPS (Fig. 4d, e). The transferred NOEs (tr-NOE) appeared in the LPS-targeting domain and the killing domain for SCPs-A6 and G6. In the $^1$H spectra, an NMR signal-broadening effect could also be observed after the addition of LPS to peptides (Supplementary Fig. 13). These experiments indicated that SCPs-A6 and G6 can interact with LPS.

**SCPs prevent LPS binding to LBP and the cell surface**. The LPS-immobilized plate was used to detect the effects of peptides on the ability of LPS to bind LBP, which can catalyze the transfer of LPS to CD14. The binding of LPS to LBP was significantly inhibited by the peptides and PMB in a concentration-dependent manner (Fig. 5a). The LPS-LBP binding rates were reduced to

**Table 2 MIC values of the SCPs, PMB, and control peptides.**

| Strains | MIC | | | | | | | | | | | | | | | | | |
|---|---|---|---|---|---|---|---|---|---|---|---|---|---|---|---|---|---|---|
| | A6 | | G6 | | N6 | | PMB | | LBP14 | | LBPN6 | | N6CK | | A6CK | | G6CK | |
| | μg per ml | μM | μg per ml | μM | μg per ml | μM | μg per ml | μM | μg per ml | μM | μg per ml | μM | μg per ml | μM | μg per ml | μM | μg per ml | μM |
| **Gram-negative bacteria** | | | | | | | | | | | | | | | | | | |
| Escherichia coli CVCC195[a] | 8 | 1.5 | 4 | 0.88 | 1 | 0.4 | 1 | 0.84 | 8 | 4.53 | 64 | 15.2 | >64 | >25.8 | 64 | 12.4 | 32 | 7.1 |
| E. coli CVCC1515[a] | 16 | 3 | 16 | 3.52 | 1 | 0.4 | 1 | 0.84 | 8 | 4.53 | 32 | 7.6 | >64 | >25.8 | >64 | >12.4 | 16 | 3.5 |
| Salmonella typhimurium ATCC14028[c] | 8 | 1.5 | 4 | 0.88 | 4 | 1.6 | 2 | 1.68 | 16 | 9.06 | 32 | 7.6 | >64 | >25.8 | >64 | >12.4 | 32 | 7.1 |
| S. pullorum CVCC533[a] | 16 | 3 | 16 | 3.52 | 2 | 0.8 | 1 | 0.84 | 32 | 18.1 | 32 | 7.6 | >64 | >25.8 | >64 | >12.4 | >64 | >14.2 |
| S. pullorum CVCC1809[a] | 8 | 1.5 | 8 | 1.76 | 2 | 0.8 | 1 | 0.84 | 16 | 9.06 | 32 | 7.6 | >64 | >25.8 | >64 | >12.4 | 32 | 7.1 |
| Pseudomonas aeruginosa CICC10419[b] | >64 | >12 | 16 | 3.52 | 16 | 6.4 | >64 | >53.76 | 64 | 36.2 | 64 | 15.2 | >64 | >25.8 | >64 | >12.4 | >64 | >14.2 |
| **Gram-positive bacteria** | | | | | | | | | | | | | | | | | | |
| Staphylococcus aureus ATCC43300[c] | >64 | >12 | 16 | 3.52 | 16 | 6.4 | >64 | >53.76 | 64 | 36.2 | 64 | 15.2 | >64 | >25.8 | >64 | >12.4 | >64 | >14.2 |
| Enterococcus faecalis CVCC3936[a] | 32 | 6 | 16 | 3.52 | 32 | 12.8 | >64 | >53.76 | 32 | 18.1 | 8 | 1.9 | >64 | >25.8 | >64 | >12.4 | 16 | 3.5 |
| Streptococcus suis CVCC606[a] | 32 | 6 | 16 | 3.52 | 64 | 25.6 | >64 | >53.76 | >64 | >36.2 | 32 | 7.6 | >64 | >25.8 | >64 | >12.4 | >64 | >14.2 |
| Bacillus subtilis ATCC6633[c] | 8 | 1.5 | 8 | 1.76 | 1 | 0.4 | 0.5 | 0.42 | 32 | 18.1 | 16 | 3.8 | >64 | >25.8 | 16 | 3.1 | 4 | 0.9 |
| Listeria ivanovii ATCC19119[c] | >64 | >12 | >64 | >14.08 | >64 | >25.6 | >64 | >53.76 | 8 | 4.53 | 8 | 1.9 | >64 | >25.8 | >64 | >12.4 | 8 | 1.8 |
| **Fungi** | | | | | | | | | | | | | | | | | | |
| Candida albicans CGMCC2.2411[d] | >64 | >12 | >64 | >14.08 | >64 | >25.6 | >64 | >53.76 | >64 | >36.2 | >64 | >15.2 | >64 | >25.8 | >64 | >12.4 | >64 | >14.2 |

Data are representative of three independent experiments. The MDR E. coli CVCC195 strain is resistant to tetracyclines (doxycycline), penicillins (benzocillin and penicillin), lincosamides (lincomycin and clindamycin), glycopeptides (vancomycin), amphenicols (chloramphenicol), macrolipids (erythromycin), aminoglycosides (amikacin, streptomycin and gentamicin) and sulfonamides (sulfamethoxazole), respectively.
PMB: polymyxin B; NN: no detection, N6CK: N6 scramble control, A6CK: A6 scramble control, G6CK: G6 scramble control.
[a]China Veterinary Culture Collection Center (CVCC).
[b]China Center of Industrial Culture Collection (CICC).
[c]American Type Culture Collection (ATCC).
[d]China General Microbiological Culture Collection Center (CGMCC).

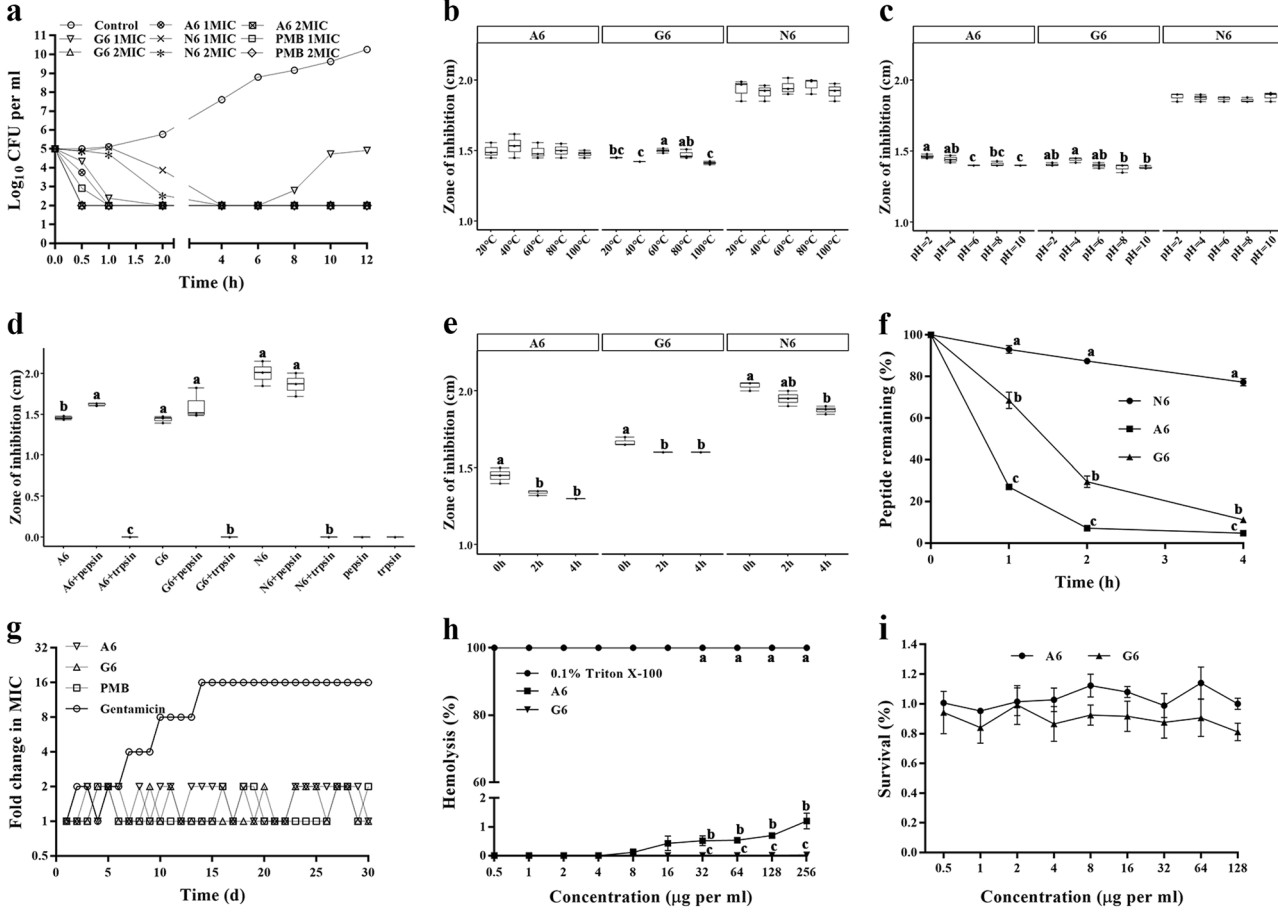

**Fig. 2 Time-killing curves, stability, and resistance of SCPs-A6 and G6 against MDR *E. coli* CVCC195 and their toxicity toward eukaryotic cells. a** The time-killing curves of A6, G6 and N6 against *E. coli* CVCC195 in vitro. Control: PBS. PMB: polymyxin B. **b–e** Effects of temperature (**b**), pH (c), enzyme (**d**), serum (**e**) on the antibacterial activity of A6, G6, and N6 against *E. coli* CVCC195. **f** The peptide remaining in serum. **g** Resistance of A6 and G6. **h** The hemolysis of A6 and G6 against fresh mouse red blood cells. 0.1% Triton X-100 served as the positive control (100% hemolysis). **i** The cytotoxicity of A6 and G6 against RAW 264.7 monocytes. Results indicate means with SD ($n = 3$ independent experiments). Different lower case letters indicate a difference between two groups ($p < 0.05$). Source data for **a–d** can be found in Supplementary Data.

68–17% (A6) and 79–13% (G6), lower than those of N6 (82–56%) and PMB (82–22%), indicating a more potent ability of A6 and G6 to inhibit LPS-LBP interaction compared to N6 and PMB. This may be associated with an appropriate distance between the functional domains provided by the linkers, thus retaining their biological activity[35,36].

Bacterial LPS can bind to CD14 on the lipid-enriched low-density domains of the macrophage plasma membrane[37]. The binding of rhodamine-labeled SCPs or FITC-labeled LPS to RAW 264.7 cells was observed using confocal microscopy and flow cytometry, respectively. FITC-labeled LPS was localized in the RAW 264.7 cell membranes (Supplementary Fig. 14a). The incubation of RAW 264.7 cells with rhodamine-labeled peptides (0.1 μM) alone showed that A6, G6 and N6 could bind to the cell membranes (Supplementary Fig. 14a). Similar to N6 (4.87–98.5%), the binding rates of A6 and G6 were 6.1–98.6% and 3.96–98.7% at concentrations of 0.1–10 μM (Supplementary Fig. 14b), indicating that SCPs-A6 and G6 can bind to the cell membrane of macrophages in a concentration-dependent manner.

When the rhodamine-labeled SCPs were added to RAW 264.7 cells that were preincubated with FITC-LPS, A6, and G6 were colocalized together with LPS on the cell membrane. The affinity of SCPs-A6 and G6 to macrophages was higher than that of N6 (Fig. 5b), which may be ascribed to the ability of LBP14 to bind to LPS[24].

The effect of the peptides on the binding of FITC-labeled LPS to RAW 264.7 cells was detected by flow cytometry. As shown in Fig. 5c, FITC-labeled LPS could bind to the cell surface and the binding was significantly prevented by both A6 and G6, with inhibition rates of 33.2% and 44.3%, respectively. This inhibition was higher than for N6 (8%), PMB (13.5%), and LBP antibody (14.5%), indicating that SCPs-A6 and G6 can more effectively suppress the binding of LPS to macrophages than N6 alone.

LPS-activated macrophages are used to evaluate the anti-inflammatory effects of drugs by using enzyme-linked immuno-sorbent assay (ELISA)[37]. 0.1 μg per ml LPS alone induced the production of tumor necrosis factor-α (TNF-α), interleukin-6 (IL-6), and interleukin-10 (IL-10) to ~1065, 558, and 246 pg per ml, respectively (Supplementary Fig. 15). After treatment with 20 μM A6, G6, N6, and PMB, the expression levels of TNF-α, IL-6, and IL-10 decreased by 518–654, 246–311, and 124–156 pg per ml, respectively, with inhibition rates of 48.6–61.4% (TNF-α), 44.1–55.7% (IL-6), and 50.4–63.4% (IL-10). For 1 μg per ml LPS, the inhibitory rates of A6, G6, N6, and PMB were 50–70.2% (TNF-α), 43.2–65.1% (IL-6), and 48.4–72.2% (IL-10). A6 more potently blocked the cytokine production than G6 or N6. This suggests that both A6 and G6 significantly inhibit LPS-induced cytokine production, which may be due to the potent ability of LBP14 to bind to LPS and further inactivate the related downstream mitogen-activated protein kinase (MAPK) cascade[38].

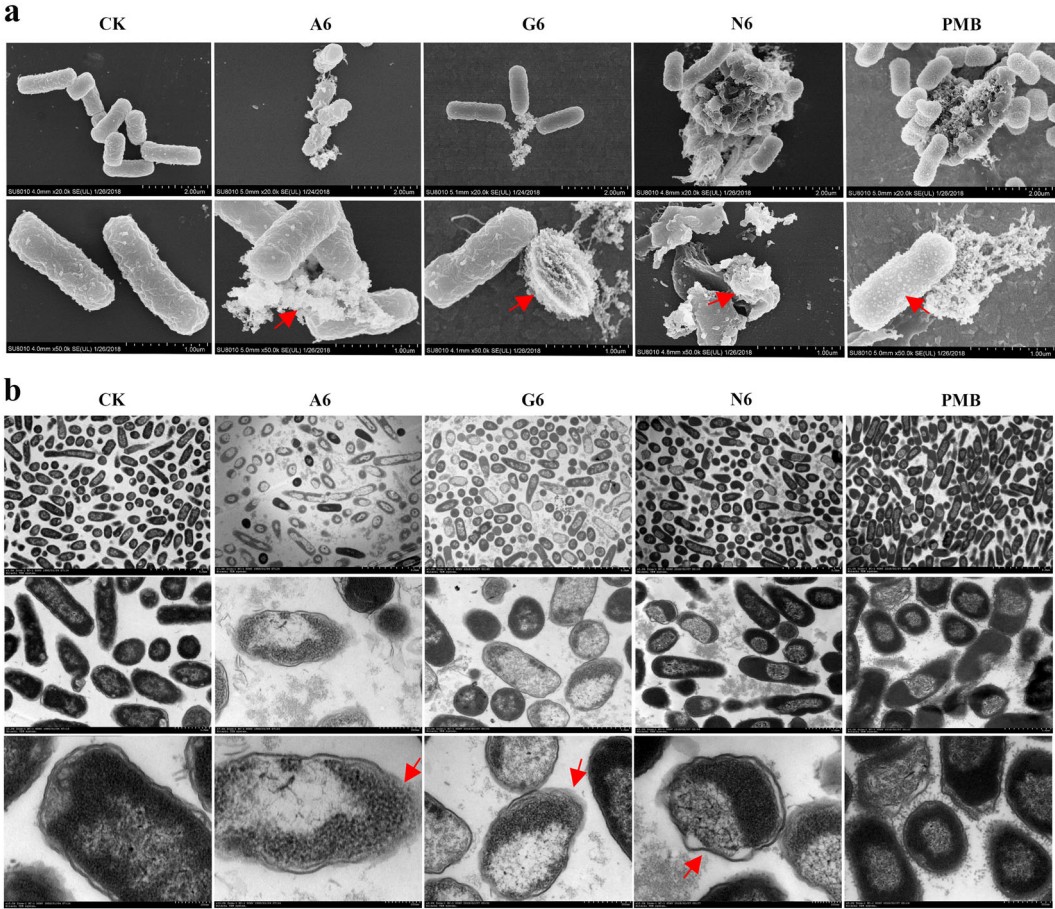

**Fig. 3 Effects of SCPs-A6 and G6 on the cell morphologies and ultrastructures of MDR *E. coli* CVCC195.** Bacteria in mid-logarithmic growth were treated with peptides or antibiotic at 4 × MIC for 2 h (*n* = 3 independent experiments). **a** SEM images. The scalebar represents 2 and 1 μm. **b** TEM images. Both N6 and PMB were used as controls. Red arrows indicated typical disruptions, which was caused by peptides (filamentous substances, disappearance of membranes, leakage of contents and ghosts) or PMB (protrusions). The scalebar represents 5, 1, and 0.2 μm.

**The SCPs protect mice from lethal challenge with *E. coli* or LPS.** After intraperitoneal injection of FITC-labeled peptides or free FITC, in the control group, free FITC was rapidly distributed and slowly cleared from the mouse body (Fig. 5d; Supplementary Table 4). At 0.5 h after injection, N6 slowly spread to lungs over time until 4 h, and was excreted 72 h post injection. Comparably, however, the A6 and G6 fluorescence was rapidly distributed throughout the body at 0.5 h post injection; it decreased after 2 h and disappeared after 48 h, indicating a more extensive distribution and the rapid metabolism of the SCPs compared to N6.

To investigate the in vivo efficacy of the SCPs, mice were intraperitoneally injected with MDR *E. coli* (LD$_{100}$ of 5 × 10$^8$ CFU) or LPS (LD$_{100}$ of 13.5 mg per kg), followed by treatment with peptides or antibiotic. The untreated mice in the control group died within 2 d (for *E. coli*) and 4 d (for LPS) (Fig. 6a). In the *E. coli* inoculation group, the survival rates were 60–80% and 80–100% after treatment with 2–4 μmol per kg A6 or G6, identical or superior to N6 (40–80%). After treatment with 0.125–0.25 μmol per kg PMB, the survival rates of the mice were 20–100%.

For LPS injection, after treatment with 0.125 μmol per kg A6 or G6, the mouse survival was 100%, which was higher than that of N6 (60%) (Fig. 6a). The survival rates of mice treated with 0.0625 μmol per kg A6 or G6 were 60% and 80%, respectively. All mice survived at doses of 0.25 μmol per kg N6 and 10 μmol per kg PMB. These results indicate that both A6 and G6 are more effective than N6 and PMB against LPS in mice. Bacterial LPS is one of the most important factors that initiates cytokine release[36].

The effects of peptides on cytokines were determined by ELISA. A significant increase in TNF-α (348.3 pg per ml) and IL-6 (538.3 pg per ml) was observed in LPS-injected mice, as compared to a blank control group (Fig. 6b). At a dose of 0.25 μmol per kg, there was a 37.3–50.1% reduction in TNF-α production and a 2.9–8.5% increase in IL-6 production in serum taken 2 h after treatment with A6, G6 and N6. Comparably, after treatment with 10 μmol per kg PMB, the TNF-α and IL-6 levels were increased by 8.2% and 18.1%, respectively. Moreover, the levels of anti-inflammatory cytokine IL-10 (39.5%) in mice decreased dramatically after LPS-stimulation; A6 and G6 more markedly increased the IL-10 level than did N6 (20.0–29.2%). These data indicate that SCPs-A6 and G6 can dually regulate LPS-induced cytokines.

Intestinal alkaline phosphatase (IAP) can dephosphorylate LPS and thus relieve the LPS toxicity[39]. The effects of the SCPs on the IAP level were shown in Fig. 6b; the IAP level was reduced by 27.6% when mice were exposed to LPS for 2 h. After treatment with peptides, the IAP expression level was markedly promoted by 15.5% (A6) and 11.5% (G6). This was higher than that promoted by N6 (4.5%), but lower than that by PMB (27.1%). These results indicate that the SCPs can better relieve the LPS toxicity than N6 by increasing IAP expression due to the LBP14 addition.

To investigate whether the SCPs protect mice from LPS-induced lung injury, the lungs were dissected and examined at 1 d and 6 d after treatment with peptides. The lungs from mice injected with LPS alone were significantly swollen and damaged,

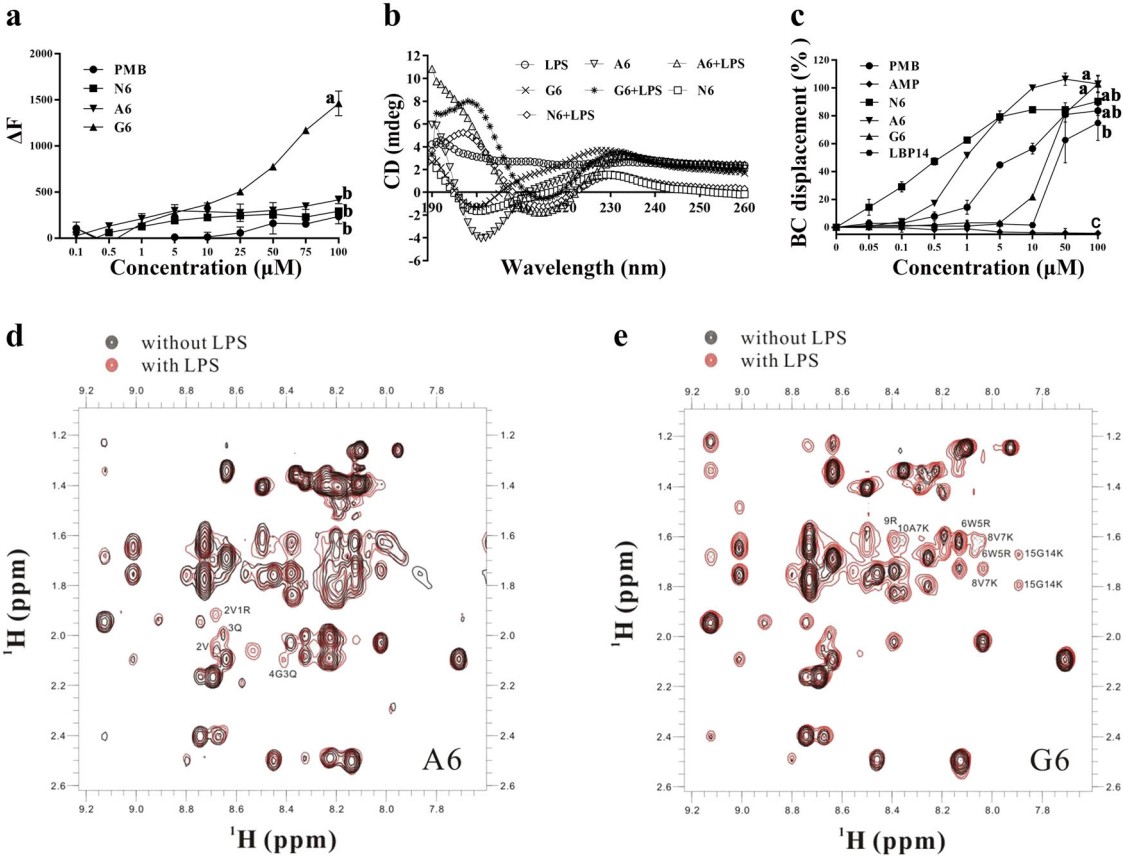

**Fig. 4 Interaction between SCPs-A6 or G6 and LPS. a** Dissociation of LPS aggregates. FITC-labeled LPS micells were incubated with A6, G6, N6, and PMB, and the fluorescent intensity was detected in a microplate reader. Results indicate means with SD ($n = 3$ independent experiments). **b** CD spectra for A6, G6, or N6 with or without *E. coli* LPS (0.2 mg per ml). **c** Binding affinity of LBP14, A6, G6, and N6 to LPS. Ampicillin (AMP) and PMB were used as negative and positive controls, respectively. Results indicate means with SD ($n = 3$ independent experiments). Different lower case letters indicate a difference between two groups ($p < 0.05$). Source data for **a**–**c** can be found in Supplementary Data 1. **d**, **e** NMR analysis of A6 (**d**) and G6 (**e**) binding to LPS. Part of the NOESY spectra for A6 and G6 showed an NOE enhancement upon addition of LPS to the peptide solvent (black: free peptide; red: peptide with LPS). Some of the tr-NOEs for the targeting domain (residues 1–14 in A6 and G6) showed that intra-residue and/or sequential NOEs were labeled beside the peaks.

with non-uniformly distributed alveoli and a large number of inflammatory cells (Fig. 6c). There was a partial alleviation of tissue swelling and injury at 1 d after treatment with 0.25 μmol per kg A6, G6 or N6; no pathological changes were found in the lung tissues at 6 d. The efficacies of SCPs-A6 and G6 were higher than those of N6 or PMB, suggesting that the SCPs improve the LPS-induced lung damage in mice.

The effects of the SCPs on LPS-induced signal pathways involved in inflammation were detected by western blotting. As shown in Fig. 7 and Supplementary Fig. 16, 13.5 mg per kg LPS induced ERK1/2 phosphorylation in the cytosol and the nuclear factor kappa-B (NF-κB) p-p65 subunit translocation from cytoplasm into nucleus, but not the translocation of inhibitor of kappa B alpha (IκBα). A6 (89.7% inhibition) and PMB (98.4%) significantly disturbed the LPS-induced translocation of the NF-κB p-p65 subunit from the cytoplasm to the nucleus, higher than for G6 (25.3%) and N6 (5%). ERK1/2 phosphorylation was markedly attenuated by G6 (88.2%) and N6 (16.1%); however, conversely, A6 and PMB moderately promoted the ERK1/2 phosphorylation, suggesting that MAPK may be involved in the enhancement of cytokine release. Similar to N6 and PMB, A6 and G6 promoted LPS-induced IκBα levels. These results demonstrate that SCPs-A6 and G6 exert their anti-inflammatory effects through the selective inhibition of the MAPK and NF-κB inflammatory signaling pathways.

## Discussion

Diarrheal diseases induced by Gram-negative bacteria such as MDR *E. coli* remain one of the major severe health threats[1]. LPS, covers >70% of the outer leaflet of Gram-negative bacteria[40], is spontaneously released from the outer membrane during bacterial growth or after exposure to antibiotics, and it is responsible for sepsis[41]. Thus, LPS can be a valid drug target to develop anti-endotoxic and antimicrobial compounds. In this study, SCPs-A6 and G6 were designed by connecting LBP14 (a LPS-targeting domain) with N6 (a killing domain) via linkers, to retain the desired bioactivity of each domain.

In the absence of linkers, direct fusion of functional domains may cause misfolding of CPs and impaired biofunction due to the limited spatial distance;[22,42,43] thus, a proper linker is needed to connect the two functional domains to reduce their interference with each other, improve folding and retain the domains' bioactivity[16,17]. Rigid and flexible linkers have been successfully applied to construct some CPs or proteins, such as Syn-GNU7, LHP7, and LL-37-haFGF[16,20,44]. Similarly, in our study, the higher MIC values of LBPN6 without a linker compared to SCPs indicated its very weak antibacterial activity, which may be related to a relatively limited spatial distance between the two domains (Table 2; Supplementary Fig. 3). The CD spectra and NMR analysis of the SCPs showed antiparallel strands, α-helices and β-turns in the killing domains in A6 and G6, similar to N6 (Fig. 4b;

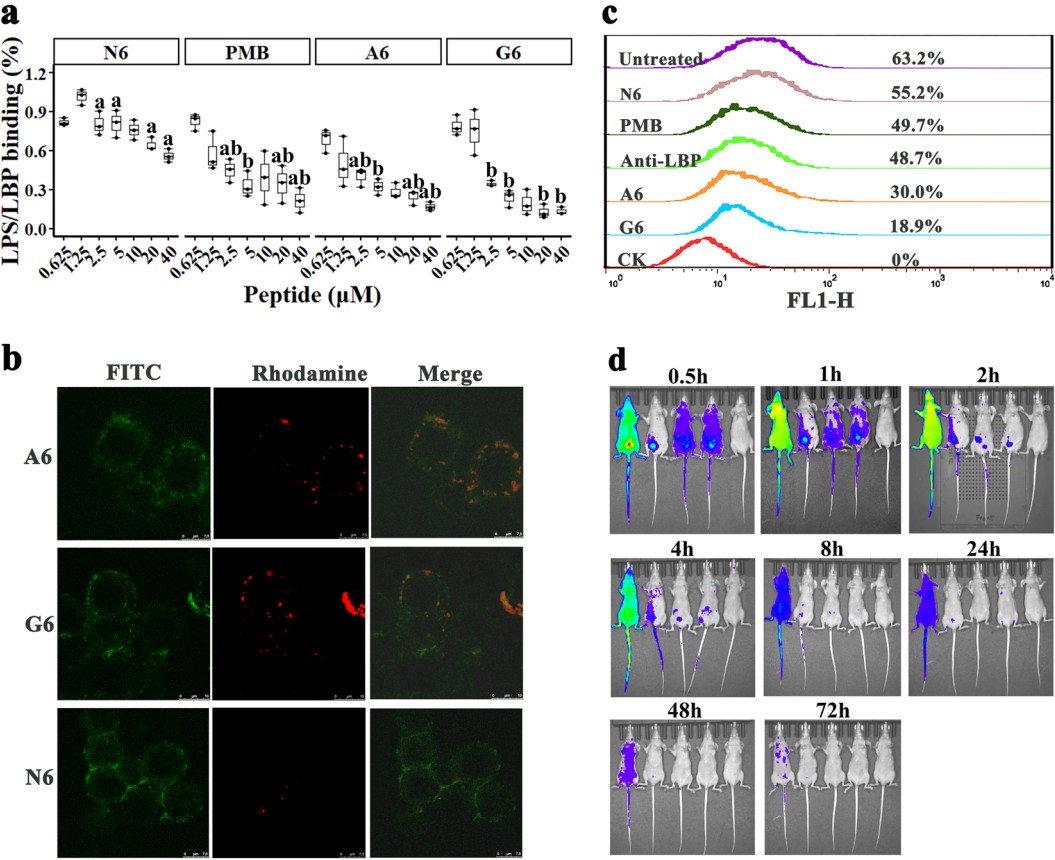

**Fig. 5 Effects of SCPs-A6 and G6 on LPS binding to LBP and macrophages. a** Effects of A6 and G6 on the binding of LPS and LBP. The 96-well microtiter plates were coated with 4 ng per ml LPS overnight at 4 °C, washed three times with PBST, and blocked with PBS for 2 h. After washing again, the plates were incubated with different concentrations of A6 and G6 at 37 °C for 1 h. After the addition of primary and secondary antibodies, the absorbance was measured at $OD_{450\,nm}$. N6 and PMB were used as controls. Results indicate means with SD ($n = 3$ independent experiments). Different lower case letters indicate a difference between two groups ($p < 0.05$). **b** Location of A6 and G6 in RAW 264.7 macrophages. Macrophages were preincubated with FITC-labeled LPS (100 μg per ml), washed, treated with rhodamine-labeled A6 and G6 (0.1 μM), and analyzed by CLSM. The scalebar indicates 7.5 and 10 μM. **c** Effects of A6 and G6 on the binding of LPS to RAW 264.7 cells. Cells were cultured overnight and incubated for 30 min at 37 °C with FITC-labeled LPS (100 μg per ml), followed by washing with PBS. Both A6 and G6 (0.1 μM) were added into cells, incubated for 1 h and analyzed by flow cytometry. N6 and PMB were used as controls. **d** Biodistribution of A6 and G6 in the healthy nude mice. The nude mice were injected intraperitoneally with 5 mg per kg FITC-labeled A6, G6, N6 or free FITC, and fluorescence (ventral) was observed at 0.5, 1, 2, 4, 8, 24, 48, and 72 h, respectively. The mice from left to right were free FITC, FITC-labeled N6, FITC-labeled A6, FITC-labeled G6 and the blank control, respectively. Source data for **a** can be found in Supplementary Data 1.

Supplementary Fig. 3). This indicated that both linkers could maintain a certain distance between the domains, which contributed to the retention of their independent structures and biological activities[17]. The bactericidal kinetics and EM images showed that both A6 and G6 more rapidly and selectively killed *E. coli* and more seriously damaged bacteria than N6 alone, and even PMB (Figs. 2a and 3; Supplementary Fig. 6), which may be related to the increased interaction between SCPs and bacteria through binding to LPS[16]. Moreover, higher in vitro binding ability of A6 and G6 to LPS than N6 (Fig. 4c) is consistent with their higher affinity to LPS or lipid A (Supplementary Figs. 11, 12 and Table 3), which may be ascribed to the LBP14 attachment to N6 and thus enhance its LPS-binding ability[24]. It has been demonstrated that Arg-Trp-Lys in LBP14, a BZB motif (B: basic amino acids, Z: arbitrary ones), is often observed in some LPS-binding molecules[45], suggesting that A6 and G6 are likely to have more LPS-binding sites than N6. This was also precisely confirmed by MD and NMR analysis of the peptide-LPS interaction (Fig. 4d and e; Supplementary Fig. 9), indicating that SCPs have stronger LPS-binding abilities than N6[25]. Moreover, SCPs-A6 and G6 had higher LPS-neutralizing and antibacterial activity against

MDR *E. coli* than N6 alone or PMB in mice (Fig. 6a). This prominent enhanced dual-function of A6 and G6 can likely be attributed to the linkers that providing a spatially-appropriate arrangement for more efficient interactions between peptides and bacteria (Supplementary Figs. 3b, 4 and 5). Notably, MDR *E. coli* CVCC195 did not develop resistance to SCPs-A6 or G6 after 30 passages, and they displayed low toxicity (Fig. 2g, h and i), which is superior to other CPs such as Syn-GNU7[16].

LPS molecules tend to form more active multimeric micelles[46]. In this study, a more significant increase in the fluorescence intensity of FITC-LPS multimers occurred after the addition of SCPs to the LPS micelles (Fig. 4a), indicating that SCPs-G6 and A6 have a stronger ability to depolymerize LPS than does N6 or PMB (Fig. 4a), leading to destabilization of the LPS assembly and a reduction in toxicity;[47] this may be related to the ability of LBP14 to bind to LPS and depolymerize LPS polymers into monomers[48]. Other pardaxins, magainin and LL-37 also had a similar effect on the LPS structures[49]. Moreover, similar to human cathelicidin CAP18 and guinea pig CAP11[50,51], A6 and G6 inhibited the binding of LPS to LBP (Fig. 5a), and therefore suppressed the LPS-LBP complex binding to macrophages,

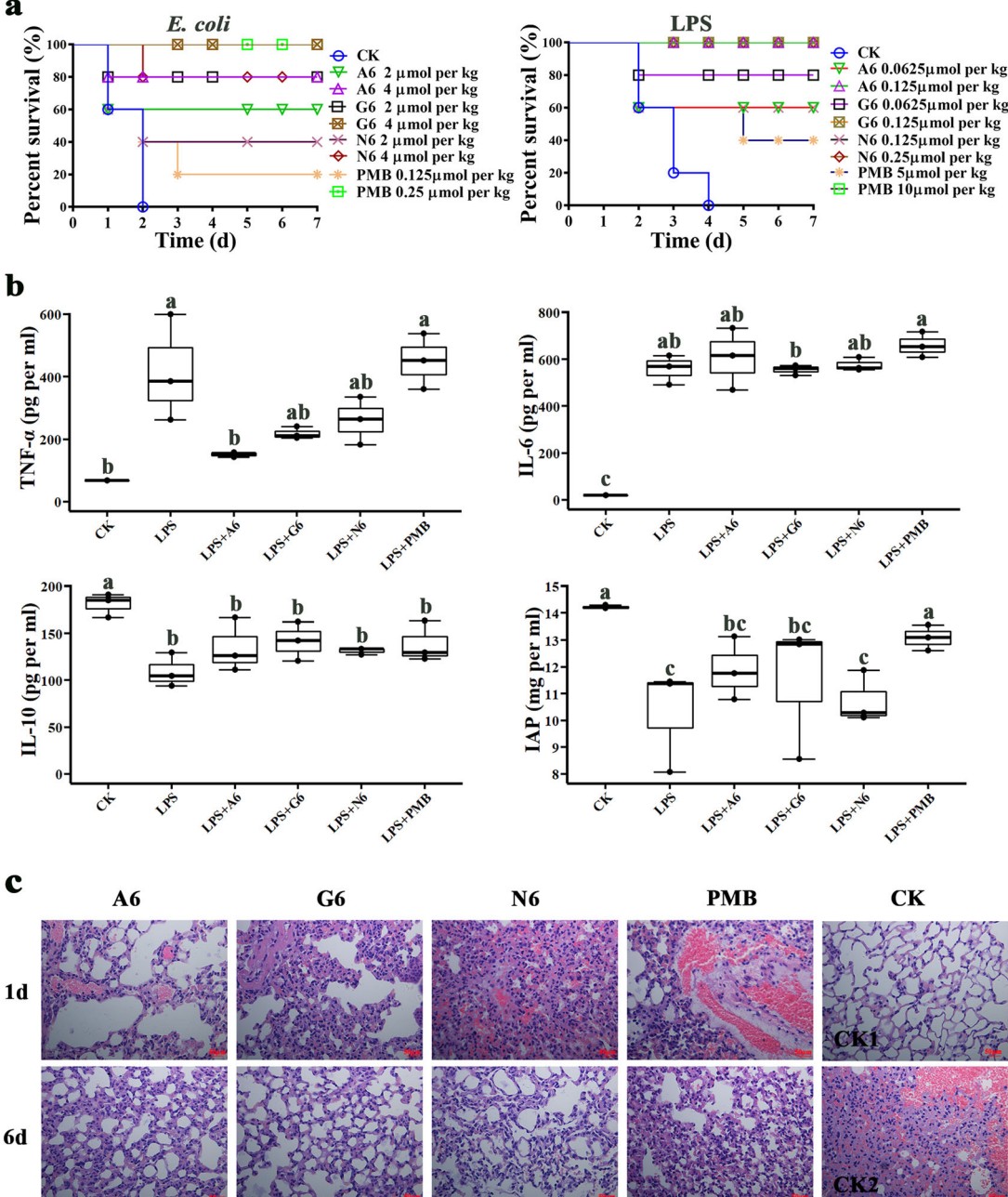

**Fig. 6 Efficacy of SCPs-A6 and G6 in mice challenged with MDR *E. coli* and LPS. a** Survival of mice. Mice were intraperitoneally injected with *E. coli* (5 × 10⁸ CFU) or LPS (13.5 mg per kg of body weight), followed by injection with A6 and G6 at 0.5 and 8 h, respectively. N6 and PMB were used as controls. The mouse survival was recorded for 7 d ($n = 5$ independent mice). **b** Effects of A6 and G6 on the cytokines and IAP levels. The results are given as the mean ± SD ($n = 3$ independent experiments). Different lower case letters indicate a difference between two groups ($p < 0.05$). **c** Effects of A6 and G6 on lung injuries induced by LPS. Mice were injected intraperitoneally with LPS and were treated with A6, G6, N6 (0.25 μmol per kg), and PMB (10 μmol per kg), respectively. Lungs were harvested and detected at 1 d and 6 d post infection. CK1: the mice unjected with LPS; CK2: the LPS-injected mice without treatment. The scalebar indicates 50 μm. Source data for **a** and **b** can be found in Supplementary Data 1.

effectively reducing the probability of LPS-mediated triggering of inflammation in hosts[52]. Additionally, A6 and G6 bound to the RAW 264.7 cell surface (Fig. 5b; Supplementary Fig. 14), which inhibited the LPS binding to macrophages and suppressed the cytokine expression (Supplementary Fig. 15)[50]. This is different from other peptides such as LL-37 that could translocate into eukaryocytes by endocytosis, accumulate in the perinuclear regions and display activity within cells, which may be associated to the cell types and peptides[53].

Host cells can recognize and respond to LPS via the signaling cascade that results in the MAPKs activation, which is involved in the production of inflammatory mediators and cytokines[54]. In this study, SCPs-A6 and G6 downregulated the inflammatory cytokines in macrophages and in mice (Fig. 6b; Supplementary Fig. 15), which is consistent with a previous report that found LL-37 to suppress pro-inflammatory cytokines in response to LPS and prevented the macrophage activation[55]. Additionally, A6 and G6 upregulated the IL-10 and IAP levels in mice challenged with

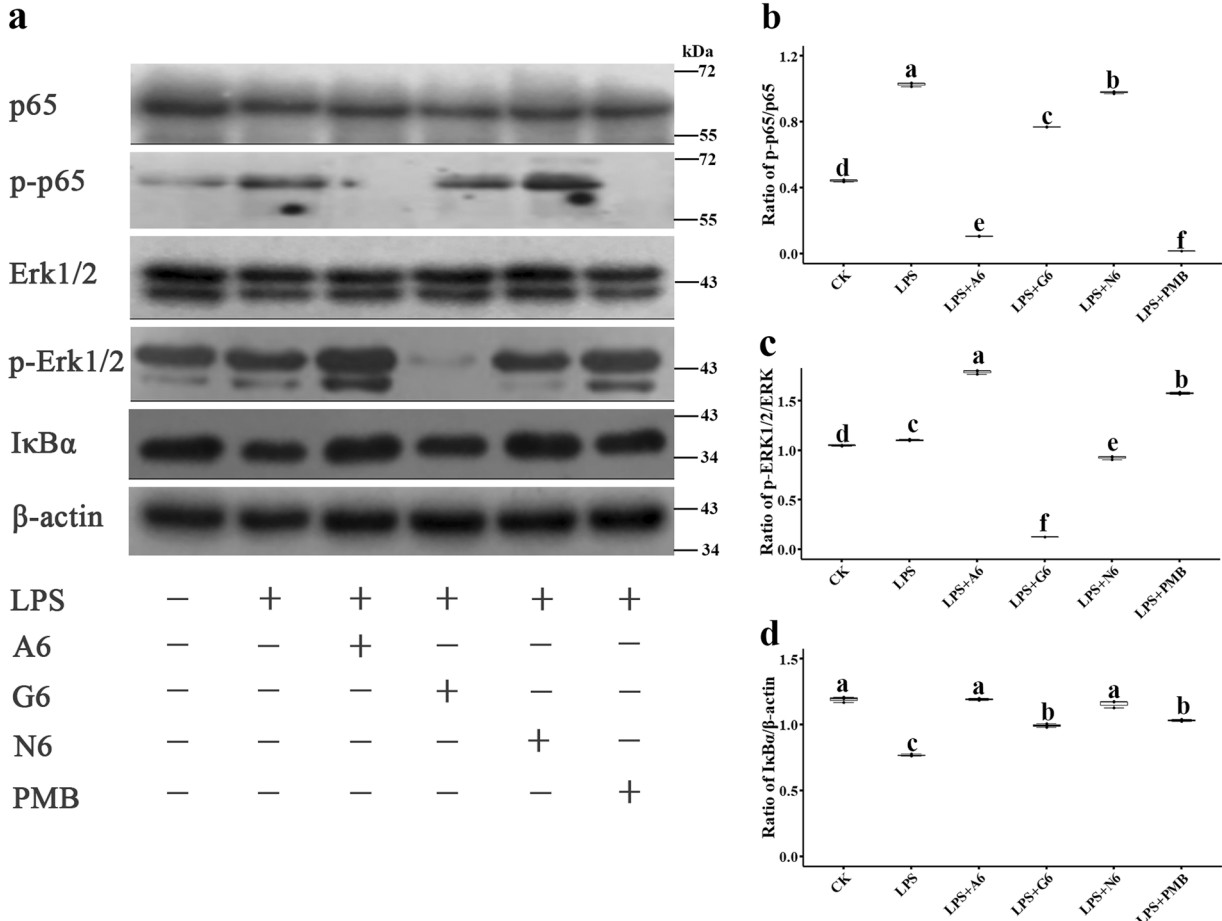

**Fig. 7 Effects of SCPs-A6 and G6 on LPS-induced NF-kB and MAPK signaling pathways in lung tissues.** The mice were injected with LPS and treated with A6, G6, N6, and PMB. **a** The protein levels of p65, p-p65, ERK1/2, p-ERK1/2, and IκBα in lungs were analyzed by western blotting. **b** Densitometric analysis of p-p65/p65 ratio. **c** Densitometric analysis of p-ERK1/2/ERK1/2 ratio. **d** Densitometric analysis of IκBα/β-actin ratio. Results indicate means with SD ($n = 3$ independent experiments). Different lower case letters indicate a difference between two groups ($p < 0.05$). Source data for **b–d** can be found in Supplementary Data 1.

LPS (Fig. 6b). IAP has an anti-inflammatory effect due to its LPS detoxification, and it can repress the downstream toll-like receptor (TLR)-4- and MyD88-dependent inflammatory cascades;[49,55] meanwhile, IAP can block p65 dephosphorylation and thus prevent NF-κB translocation to the nucleus, thereby downregulating the pro-inflammatory responses[56]. The SCPs more promoted LPS-induced IκBα levels and reduced NF-κB p65 phosphorylation than N6 alone. The phosphorylation of ERK1/2 was decreased by G6 and N6, but not by A6 or PMB (Fig. 7). These results imply that the anti-inflammatory activity of G6 and N6 is related to the increased IAP levels, which may be regulated through the inhibition of NF-κB activation and blocking of the ERK1/2 pathways. The SCPs also alleviated organ injury and enhanced in vivo detoxification activity more effectively than N6 or PMB (Fig. 6), which may be candidates for treating LPS-induced endotoxemia.

In summary, SCPs were successfully designed by adding LPS-targeting peptide to a killing peptide via linkers. They displayed higher LPS-neutralizing and killing abilities against MDR *E. coli* than their parents, including disrupting bacterial membranes, binding to LPS, inhibiting LPS binding to LBP and macrophages, suppressing inflammatory cytokines and promoting IAP levels by NF-kB and MAPK signaling pathways, alleviating lung damage and protecting mice challenged with *E. coli* and LPS (Fig. 8). The SCPs did not induce bacterial resistance and had low toxicity. These results suggest that the SCPs may be promising dual-function

candidates as improved antibacterial and anti-endotoxin agents to treat MDR *E. coli* and sepsis; it provides new clues for the design of dual-functional chimeric peptides by different linkers and gives the preliminary data support for preclinical/clinical studies.

## Methods

**Synthesis and property/structure analysis of peptides.** All peptides, including the SCPs and their scrambles, were synthesized by Mimotopes (Wuxi, China), and their purity was greater than 90%. The physicochemical properties (molecular weight, isoelectric point, net charge, and aliphatic index) of A6, G6, and their parents were calculated by ProtParam (https://web.expasy.org/protparam/). The Boman index, hydrophobicity and alpha helix content of the peptides were analyzed, predicted and calculated by the Antimicrobial Peptide Calculator and Predictor (http://aps.unmc.edu/AP/prediction/prediction_main.php; http://npsa-pbil.ibcp.fr/cgi-bin/npsa_automat.pl?page=npsa_hnn.html),
 AntiBP2 (http://crdd.osdd.net/raghava/antibp2/submit.html), and HeliQuest (http://heliquest.ipmc.cnrs.fr/cgi-bin/ComputParamsV3.py), respectively.

**NMR analysis and structure calculation of peptides.** The structures of the SCPs in aqueous solvent were analyzed by solution state NMR at 298 K on an Agilent DD2 600 MHz spectrometer equipped with a cold probe. For each peptide, 5 mg was dissolved in 500 μl of aqueous solvent (10% $D_2O$, 90% $H_2O$ V/V) to make the NMR samples. A series of 1D and 2D spectra, including $^1H$, $^1H$-$^1H$ TOCSY, $^1H$-$^1H$ NOESY, $^1H$-$^{13}C$ HSQC, and $^1H$-$^{15}N$ HSQC, were acquired for resonance assignment and structure analysis. All the TOCSY experiments were carried out using an 80 ms MLEV-17 spin-lock with a field strength of 6839 Hz. The mixing time for each NOESY experiment was 150 ms. The spectra widths were 7396.4 × 7396.4 Hz for $^1H$–$^1H$, 7396.4 × 21124.9 Hz for $^1H$–$^{13}C$, and 7396.4 × 1945.6 Hz for the $^1H$-$^{15}N$ correlation spectra. A total of 256, 140, and 128 complex points were collected

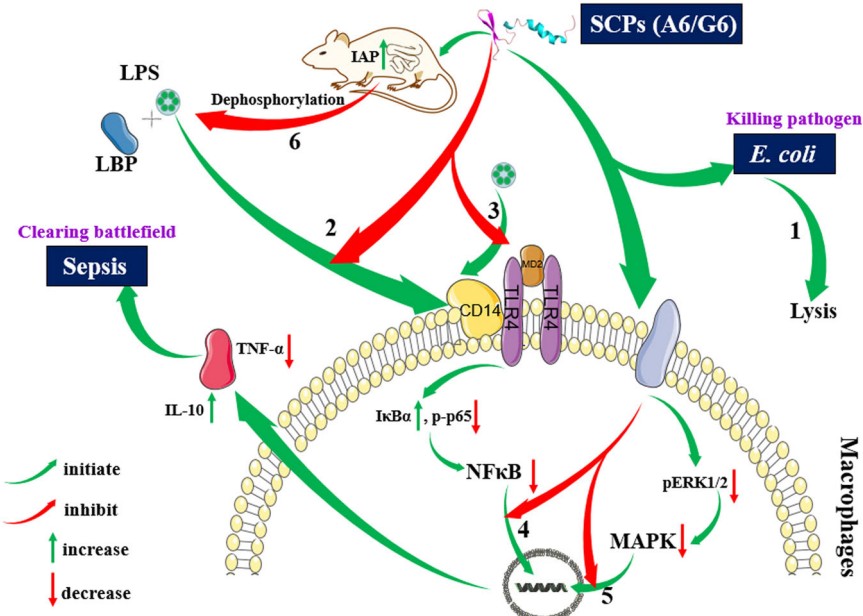

**Fig. 8 An outline of SCPs-A6 and G6 in combating bacterial infection and preventing inflammation.** Dual-functional A6 and G6 can directly kill *E. coli* either by disruption of cell membranes or internal biomacromolecules (1), directly bind to LPS or macrophages (2), which may be compete with LPS for binding to the TLR signaling complex (3), suppress NF-κβ translocation into the nucleus (4), regulate the inflammatory cytokines by the MAPK pathway (5), improve the IAP levels in the duodenum of mice and thus indirectly dephosphorylate LPS (6) and relieve its toxicity[6,37]. Images of LBP, cytokines, CD14, MD2, TLR4, and membrane proteins, partial cell membranes, cell membranes, and nucleic acids were from SMART—Servier Medical ART (http://smart.servier.com/), which is licensed under a Creative Commons Attribution 3.0 International License (https://creativecommons.org/licenses/by/3.0/).

for the indirect proton, carbon and nitrogen dimensions, respectively, and 512 complex points were collected for the direct observation dimension. The chemical shifts were referenced to the DSS at 0.00 ppm for the proton and indirectly referenced for the carbon and nitrogen dimension[57]. All NMR data were processed using NMRPipe and analyzed with the CcpNmr program suite[58]. Signal assignment and inter-proton distance restraints were derived from the NOESY spectrum. Backbone dihedral angle restraints (Φ and Ψ angles) were derived using the program DANGLE incorporated in the CcpNmr suite, and thus the secondary structure prediction was made. Structure calculation and NOE assignment were performed simultaneously by using the program CNS and ARIA2[59,60]. A total of 100 structures were calculated and 20 (for A6) or 10 (for G6) structures with the lowest total energy were selected to perform a refinement procedure in water. The atomic coordinates of the peptides and NMR data had been deposited in the Protein Data Bank and the Biological Magnetic Resonance Data Bank (see 'data availability' for detail). The dihedral angle distribution was analyzed by Procheck-NMR[61] and the ramachandran statistics for the ordered residues of A6 (residues 3–23 and 29–43) in the most favoured regions, additionally allow regions, generously allowed regions and disallowed regions were 88.7%, 11.3%, 0.0%, and 0.0%, respectively. And those for G6 (residues 9–12 and 24–38) were 88.3%, 11.7%, 0.0%, and 0.0%, respectively.

**Antibacterial activities and bactericidal curves of the SCPs.** The MIC values of A6, G6, the parental peptides (N6 and LBP14), LBPN6, scramble controls (N6CK, A6CK, and G6CK), and PMB were determined by the microtiter broth dilution method[25]. The tested strains included Gram-negative bacteria, Gram-positive bacteria, and fungi and were cultured to the logarithmic growth phase, diluted to $10^5$ CFU per ml, and added into 96-well plates (90 μl per well). Peptides or antibiotic were diluted with PBS by 2-fold dilution and were added to each well (10 μl per well). Polymyxin B (PMB) and PBS were used as the positive and negative control, respectively. The plates were incubated at 37 °C for 16–18 h until visible turbidity was observed in the negative control. The lowest concentration that could completely inhibit bacterial growth was the MIC value of the peptides or antibiotic against the tested strains. All experiments were repeated three times.

The bactericidal effect of peptides was investigated according to previous methods[25]. MDR *E. coli* CVCC195 cells were cultured to logarithmic phase, diluted in Mueller Hinton broth (MHB) ($1 × 10^5$ CFU per ml) containing $1 ×$ or $2 ×$ MIC of A6, G6, N6, or PMB, and cultured at 37 °C (250 rpm). Physiological saline was added as a control. Bacterial samples (100 μl) were removed at different time points (0, 0.5, 1, 2, 4, 6, 8, 10, and 12 h) and were counted on MH solid plates. The time-killing curve of A6 and G6 was plotted.

The selectivity of A6, G6, and N6 against Gram-negative and Gram-positive bacteria was determined using the mixed-culture bactericidal kinetics assay, as described in the Supplementary Information.

**Stability, resistance, and toxicity of the SCPs.** The thermal, pH and enzyme stability of A6 and G6 were evaluated against MDR *E. coli* CVCC195 using a previous method with minor revisions[25]. The N6, A6, and G6 solutions were treated at different temperatures (20–100 °C) for 1 h, dissolved and incubated at different pH values (2–10) for 3 h, or incubated with trypsin or pepsin for 4 h. The inhibition zone diameters were measured after incubation at 37 °C for 18–24 h. The peptides were dissolved in $H_2O$ or mice serum at 37 °C to detect their serum stability and samples were removed at different time points to determine their residual activity by the inhibition zone or reverse-phase high-performance liquid chromatography (RP-HPLC) methods[62,63].

Bacterial resistance against the peptides was assessed via the MIC assays[25]. Mid-log phase *E. coli* CVCC195 ($1 × 10^5$ CFU per ml) was added into 96-well plates and incubated with A6, G6 or antibiotic (from $32 ×$ to $0.25 ×$ MIC). After 16–18 h of incubation at 37 °C, cells from the second highest concentration showing visible growth were used to inoculate the subsequent cultures. The MIC values for drug passage cultures were obtained as described above. This process was repeated for 30 d.

To determine the hemolysis of the peptides, eyeball blood was collected from 6-week-old specific pathogen-free (SPF) ICR mice. The blood was centrifuged at 4 °C for 10 min (1500 rpm), washed three times with 0.9% physiological saline, and diluted to an 8% suspension. The peptides were dissolved in 0.9% physiological saline (0.5–256 μg per ml). Red blood cell suspensions and peptide solutions (100 μl) were mixed in 96-well plates, centrifuged at 4 °C for 5 min (1500 rpm), and incubated at 37 °C for 1 h. The supernatant was added into a 96-well plate, and the UV absorbance was measured at 540 nm using a microplate reader. Physiological saline and 0.1% Triton X-100 were used as 0% and 100% hemolytic controls, respectively. The hemolytic ratio was calculated as follows:[25] hemolysis (%) = [(Abs $_{540 nm peptide}$−Abs $_{540 nm saline}$)/(Abs $_{540 nm 0.1\%Triton X-100}$−Abs $_{540 nm saline}$)] × 100%.

The cytotoxicity of A6 and G6 was evaluated using the 3-(4, 5-dimethylthiazolyl-2)-2, 5-diphenyltetrazolium bromide (MTT) method[64]. RAW 264.7 cells ($2.5 × 10^5$ cells per ml) were added into a 96-well plate (100 μl per well) and cultured at 37 °C (5% $CO_2$, 95% saturated humidity) for 24 h. The medium was then removed and the cells were washed twice with PBS. The peptide solutions (from 0.5 to 128 μg per ml) were then added to the plates (100 μl per well), incubated for 24 h and washed twice with PBS. Next, 5 mg per ml MTT (100 μl per well) was added and incubated for 4 h, followed by the addition of 150 μl per well dimethyl sulfoxide (DMSO). After a few minutes of shaking in a shaker, the crystals in the bottom of wells were completely

dissolved, and the absorbance of each well was measured at 570 nm. The cell survival was calculated according to the following formula: survival (%) = (Abs $_{peptide}$/Abs $_{control}$) × 100%.

**Effects of the SCPs on the bacterial morphology**. SEM was used to examine the ultrastructural changes in bacteria treated with peptides. Peptides (A6, G6, and N6) or PMB (4 × MIC) were added into mid-log phage MDR *E. coli* CVCC 195 and *S. aureus* CVCC43300 cells and were incubated at 37 °C for 2 h. PBS was used as the negative control. After centrifugation, the bacterial precipitation was added with 2.5% glutaraldehyde and the cells were fixed at 4 °C for 24 h. After ethanol dehydration, drying and coating, samples were visualized by a QUANTA200 SEM (FEI, Philips, Netherlands)[65].

The processing and preparation of TEM samples was similar to that of SEM until glutaraldehyde fixation. After washing with PBS, bacterial cells were fixed in 1% citric acid buffer for 1 h, dehydrated in a series of acetone (50–100%) and embedded, followed by preparation into ultra-thin sections and staining with 1% uranyl acetate. The samples were observed in a JEM1400 (JEDL, Tokyo, Japan). At least 200 bacterial cells were visualized and scored for the observed characteristics[65].

**Interaction between the SCPs and bacterial LPS**. To evaluate whether peptides can depolymerize LPS multimers, 180 μl of FITC-labeled LPS (0.5 μg per ml) and 20 μl of different concentrations of peptides (A6, G6 and N6) or PMB were added into a 96-well plate, respectively. The fluorescence intensity was detected in a microplate reader ($\lambda_{excitation}$ = 490 nm, $\lambda_{emission}$ = 515 nm). FITC-labled LPS without peptides was used as a control[66].

Secondary structures of peptides (A6, G6, and N6) in *E. coli* 0111:B4 LPS solution were determined at room temperature using a Pistar π-180 spectrometer. A6 (200 μg per ml), G6 (200 μg per ml), PMB (200 μg per ml), or N6 (25 μg per ml) were mixed with LPS (400 mg per ml) (*v*:*v* = 1:1). The mixture was added into a 1-mm diameter quartz cuvette for CD scanning with a scanning range of 190–260 nm, a step resolution of 2 nm and a speed of 10 nm per min. Samples without LPS were used as controls.

MD was performed by using Autodock4.2 to predict the LPS-binding sites on A6 and G6 as described in the Supplementary Information[67]. Briefly, the SCPs (A6 and G6) and LPS were used as ligands and a receptor, respectively. A grid map (70 × 80 × 80 points) was constructed with a grid spacing of 0.375 Å and center of the glucosamine II (GlcN II) in lipid A between peptides and LPS.

The affinity of SCPs (A6, G6, and N6) to *E. coli* LPS was detected by the fluorescent BC probe[68]. The binding of fluorescent probe with LPS can cause fluorescence quenching. When peptides bind to LPS, BC probe can be replaced, which can result in an increase in fluorescence. Equal volumes of LPS (40 μg per ml) and BC probe (10 μM) in 50 mM Tris buffer (pH 7.4) were mixed and added into a black 96-well microtiter plate (180 μl mixture per well), followed by the addition of different concentrations of A6, G6, or N6 (20 μl per well). Ampicillin and PMB were used as the negative and positive control, respectively. Changes in fluorescence were measured by a fluorescence spectrophotometry at room temperature (excitation: 580 nm, emission: 620 nm) and the BC displacement rate was calculated.

The real-time interaction between ligands (A6, G6, N6, and PMB) and analytes (LPS or lipid A) was observed by the SPR technology (GE Biacore™ 8K)[69]. The immobilization of ligands onto the carboxymethylated dextran (CM5) sensor chips was carried out with A6 (20 μg per ml), G6 (20 μg per ml), N6 (10 μg per ml), or PMB (400 μg per ml) at 10 mM in sodium acetate (pH = 5.5). After immobilization, 20 mM NaOH was added into the flow cells to remove unbound ligands. The direct detection of ligands was performed for 120 s at a flow rate of 30 μl per min, followed by dissociation for 600 s. All Biacore experiments were carried out at 25 °C. The affinity of peptides or PMB to LPS or lipid A was calculated by fitting the sensorgrams of kinetic injections to the bivalent binding model with Biacore evaluation software version 3.0 and the Biacore Evaluation Software (a 1:1 binding model), respectively.

$^{1}$H and tr-NOESY were used as NMR probes to investigate the interaction between the SCPs with LPS[70]. Stock LPS solutions were titrated into peptides samples to see the line broadening effect which indicated interaction between molecules. The tr-NOESY spectra for A6 and G6 were collected with samples containing ca. 2 mM peptides and ca. 0.2 mM LPS.

**Effects of the SCPs on the binding of LPS to LBP**. The 96-well microtiter plates were coated with 4 ng per ml LPS (0.1 M Na$_2$CO$_3$ and 20 mM EDTA, pH 9.6, 50 μl per well) overnight at 4 °C, washed three times with PBST, and blocked with PBS (containing 1% BSA) for 2 h. After washing with PBST again, the plates were incubated at 37 °C for 1 h with different concentrations of peptides (A6, G6, or N6), which were dissolved in Dulbecco's Modified Eagle's (DMEM) medium (containing 10% FBS). After washing, cells were incubated with anti-LBP mAb 6G3 (ThermoFisher) for 1 h at 37 °C and HRP-conjugated rabbit anti-mouse IgG for 1 h at room temperature, respectively. Tetramethylbenzidine (TMB) solution was then added and incubated for 25–30 min. Finally, the reaction was stopped by addition of 0.5 M H$_2$SO$_4$, and the absorbance was measured at 450 nm[50].

**Location and effects of the SCPs on LPS binding to macrophages**. To determine the location of peptides in cells, mouse peritoneal macrophages (RAW 264.7 cells, $2.5 \times 10^5$ cells per well) were incubated with FITC-labeled LPS (100 μg per ml) or rhodamine-labeled A6 or G6 (0.1–10 μM) for 30 min at 37 °C and were washed with DMEM medium; the cells were then analyzed by a confocal laser scanning microscope (CLSM) and a FACS Calibur Flow Cytometer (BD, USA), respectively. Similarly, after an incubation with FITC-labeled LPS and washing with medium, cells were incubated with rhodamine-labeled A6 or G6 (0.1 μM) at 37 °C for 1 h. After washing again, the effects of peptides on the binding of FITC-labeled LPS to the cells were analyzed by CLSM and flow cytometry, respectively.

**Effects of the SCPs on cytokines in LPS-stimulated macrophages**. Macrophages RAW 264.7 were incubated with LPS (0.1 and 1 μg per ml) that was dissolved in serum-free DMEM medium, and then were treated with A6, G6, N6, or PMB (20 μM) alone in 24-well plates ($2.5 \times 10^5$ cells per well). The cells incubated with serum-free DMEM and LPS were used as a control. After treatment for 2 h, the supernatants were collected and assessed for TNF-α, IL-6, and IL-10 by using the ELISA kits in the Jiaxuan Biotech. Co. Ltd. (Beijing, China).

**In vivo experiments**. All mice used in the experiment were purchased from the Beijing Vital River Laboratory Animal Technology Co. Ltd. The animal experiments were performed according to the Animal Care and Use Committee of the Feed Research Institute of Chinese Academy of Agricultural Sciences (CAAS) and protocols were approved by the Laboratory Animal Ethical Committee and its Inspection of the Feed Research Institute of CAAS (AEC-CAAS-20090609).

**Biodistribution of the SCPs**. The nude mice were intraperitoneally injected with 5 mg per kg FITC-labeled A6, G6, and N6, respectively. Free FITC was used as a negative control. The real-time fluorescence in mice was observed at 0.5, 1, 2, 4, 8, 24, 48, and 72 h, respectively, with the excitation wavelength at 488 nm using a Maestro 2 IVIS® Spectrum CT (PerkinElmer, USA).

**A sepsis mouse model**. The 6-week-old SPF ICR mice (female, about 20 g) (5 mice per group) were intraperitoneally injected with MDR *E. coli* CVCC195 ($0.5 \times 10^7$, $0.5 \times 10^8$, and $0.5 \times 10^9$ CFU per ml) or *E. coli* LPS (9, 13.5, 18, 27, 30, and 36 mg per kg) in PBS and were continuously observed for 7 d to determine the absolute lethal dose (LD$_{100}$) in mice[25].

**Efficacy of the SCPs**. After the intraperitoneal injection with MDR *E. coli* or LPS at a concentration of LD$_{100}$, the mice were treated with a series of different concentrations of N6, A6, G6, or PMB at 0.5 h and 8 h, respectively. The mouse survival was recorded for 7 d. The mice injected with LPS or saline were used as negative or blank controls, respectively.

**Effects of the SCPs on inflammatory cytokines and IAP**. The mice were intraperitoneally injected with LPS at an absolute LD$_{100}$ and then were treated with 0.25 μmol per kg peptides (A6, G6 and N6) or 10 μmol per kg PMB at 0.5 h after challenge. The mice were sacrificed at 2 h and the sera were collected to detect inflammatory cytokines (including TNF-α, IL-6 and IL-10) using ELISA kits. To visualize IAP activity, the duodenum was rapidly excised from the mice and was homogenized; the IAP levels in the superpants were detected as described in the Supplementary Information.

**Protection of lung tissues from LPS**. The mice were treated with N6 (0.25 μmol per kg), A6 (0.25 μmol per kg), G6 (0.25 μmol per kg), and PMB (10 μmol per kg), respectively after the intraperitoneal injection of LPS at a concentration of LD$_{100}$. The lung tissues were taken at 1 d and 6 d, respectively, washed with PBS and placed in 4% paraformaldehyde for 24 h at 4 °C. After washing with PBS, the tissues were dehydrated by a series of different concentrations of ethanol (75%, 85%, 90%, and 95%) and were immersed in xylene for paraffin embedding. After sectioning, the samples were stained with hematoxylin and were observed by a light microscope. The mice challenged with LPS and PBS served as the negative and blank control, respectively.

**Effects of the SCPs on MAPK and NF-κB signal pathways**. Additionally, the mice were sacrificed at 8 h and the lung tissues were homogenized and suspended in 5 M HEPES buffer for 20 min according to standard procedures[40]. After centrifugation at 4 °C for 20 min (12,000 rpm), the protein levels of supernatants were quantified using a Bradford protein assay kit. A total of 30 μg proteins were analyzed on an 8–12% sodium dodecyl sulfate-polyacrylamide gel electrophoresis (SDS-PAGE) gel and were immunoblotted onto PVDF membranes, followed by an incubation with the primary antibodies of p65, p-p65, extracellular signal-regulated kinase (ERK)1/2, p-ERK1/2, and IκBα overnight at 4 °C. After washing, the blots were incubated with the peroxidase conjugated goat-rabbit or goat-mouse antibodies. Relative protein expression levels were quantified by a densitometric measurement of chemiluminescence (ECL) reaction band. β-actin was used as the internal reference.

**Statistics and reproducibility**. Unless otherwise noted, experiments were repeated at least three times. All data are presented as means ± standard deviation (SD). Statistical analyses between treatments or groups were determined using one-way analysis of variance (ANOVA) models in SAS 9.2 (SAS Institute Inc., Cary, NC, USA), followed by Dunnett's multiple comparisons test. A $p$-value of <0.05 was considered statistically significant.

**Reporting summary**. Further information on research design is available in the Nature Research Reporting Summary linked to this article.

## Data availability
NMR data for A6 and G6 in the PDB fromat are available wwPDB under the IDs 6K4W and 6K4V, respectively, while the NMR chemical shift data and restraint data are available in the BMRB with entries 36255 and 36254, respectively. Other data are available from the corresponding author upon request.

## Code availability
Custom codes that support the findings of this study are available from the first author upon reasonable request.

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

## Acknowledgements

This study was financially supported by National Key R & D Program of China (National Keypoint Research & Invention Program) (2018YFD0500600), the National Natural Science Foundation of China (grants No. 31772640, No. 31572444, No. 31372346, No. 31672467, and No. 31572445), the AMP Direction of National Innovation Program of Agricultural Science and Technology in CAAS (CAAS- ASTIP-2013- FRI-02), and its Key Project of Alternatives to Antibiotic for Feed Usages (CAAS-ZDXT2018008). Additionally, we acknowledge Zhao Tong from the institute of Microbiology at the Chinese Academy of Sciences (CAS) for her technical support with the flow cytometry.

## Author contributions

Z.L.W., X.M.W. and J.H.W. designed the experiments; Z.L.W and X.H.L. conducted the experiments and analyzed the NMR data; D.T., R.Y.M., Y.H., N.Y., X.W. and Z.Z.L. provided reagents and some data analysis; Z.L.W., X.M.W. and J.H.W. wrote the manuscript.

## Competing interests

The authors declare no competing interests.
