## [Peer Review File · Communications Biology]

Reviewers' comments:

Reviewer #1 (Remarks to the Author):

In their manuscript titled "Clearing the "battlefield"-lipopolysaccharide after killing multiple-d
1 rug resistant Escherichia coli by "smart" chimeric peptides-A6 and G6", Wang and coworkers build two chimeric peptides by combining the LPS targeting LBP14 with the antimicrobial N6 peptides. Two types of linkers are used to make A6 with a rigid linker or to make G6 with a flexible linker. Physicochemical properties, stability, toxicity, LPS targeting, antibacterial and antiendotoxic activities are examined. The authors suggest that A6 and N6 combine the properties of the two components and killed drug resistant E. coli and neutralized LPS toxicity. Furthermore, These dual action peptides are proposed to be novel candidates for the treatment of infection and sepsis.

Writing: The overall outline and organization of the paper is good but the text needs to be carefully edited for spelling, grammar and scientific language.

Reproducibility and transparency: The authors have performed detailed in vitro and in vivo experiments to support their claims. The level of detail in the methods section is adequate for repeating the experiments. However, the raw data is not available for the graphs. In addition, for the SEM and TEM figures, it is not clear if the selected images are representative for the whole community of cells. It would be useful to include the full scale images. It would also be useful to comment on the statistical significance of the mice experiments.

Novelty and timeliness: There are several chimeric peptides reported in the literature therefore the approach is not completely novel. Combining these two specific properties to make an antibacterial peptide with LPS neutralizing capability is a novel approach. Given the increase in antibacterial resistance and related diseases, a novel peptide design may be a significant contribution if the peptide is stable, soluble and easy to produce. The final results should be discussed in light of reports about other peptides with similar properties.

Reviewer #2 (Remarks to the Author):

Journal: Communications Biology

Manuscript #: COMMSBIO-19-0167-T

Title: Clearing the "battlefield"-lipopolysaccharide after killing multiple-drug resistant Escherichia coli by "smart" chimeric peptides-A6 and G6

Comments:

Lipopolysaccharide (LPS) is an essential component of the outer membrane of Gram-negative bacteria. which is released from the cell envelopes after destruction of the bacterial cell wall,

and always leads to inflammatory and sepsis. In this paper, Wang et al created LPS-targeted "smart" chimeric peptides (SCPs)-A6 and G6 by connecting LBP14 (targeting LPS) and N6 (killing pathogen) via rigid and flexible linkers, respectively, by which to clear "battlefield"-LPS after killing bacteria. They systematically investigated the properties, functions and mechanisms of these peptides in vitro and in vivo. The experiments were reasonably designed, and the data were well presented which clearly addressed the issue in the manuscript. As a result, these peptides seem to "be promising dual-function candidates as novel antibacterial and anti-endotoxin agents to treat bacteria and sepsis, as mentioned in the context. Therefore, it is a nice work and good data. However, the only concern to me is that the novelty of this work is limited.

1. In line 61, authors mentioned that "the activities of a few AMPs can be specifically improved by attaching a targeting region to generate novel specifically-targeted chimeric peptides that containing functionally independent targeting and killing moieties or domains", while based on the data in Table 2, peptides A6 and G6 seem no strong gram-positive and negative selectivity, even worse than N6, a killing peptide itself, please explain.

2. As compared with a similar work cited in lit. 16 [Kim, H., Jang, J. H., Kim, S. C. & Cho, J. H. Enhancement of the antimicrobial activity and selectivity of GNU7 against Gram-negative bacteria by fusion with LPS-targeting peptide. *Peptides*, 82, 60–66 (2016), what's the significance or novelty of this study?

3. From line 73 to 76, authors mentioned the differences between "flexible linker" and "rigid linker". However, through the whole manuscript, did author try a chimeric peptide that without a "linker"? is it necessary in any cases to have a linker? though authors cited three literatures in discussion (line 319-321), however, there are also a lot of studies that show good data without a linker. Actually, it seems that no significant differences in all tests between A6 and G6 within this study. Please explain.

In addition, it seems authors put too much focus on "linkers" (line 68-79, line 89-90, line 318-319, line 319-331 etc.). If so, which one confers better results in this work? Is "flexible" one or "rigid" one? Any differences, such as secondary structure in aqueous solution, in the presence and absence of LPS, LPS-neutralization ability, anti-inflammatory activity, etc. between these two peptides? The discussion may necessarily be required.

4. Line 110-111, the structures of SCPs analyzed by NMR spectroscopic technology are reasonable. However, it should be pointed out that the data were acquired in aqueous solution, while the presence of organic solvent or lipid sometimes will induce a significant change in their secondary structure. Therefore, authors have ever considered the analyses of their structures in the presence of lipid? It might be closer to the "real" situation when interacted with bacterial cell membrane.

5. Line 115-116, the phrase "which may be due to the absence of amphipathic conformations of the peptides when interacted with bacteria", please provide any possible reason in discussion. Why?

6. Line 125-130, the phrase "The result indicated that the antibacterial activity of A6 and G6 is consistent with a bactericidal mode of action." Please explain, or provide exact literature.

7. In the section of TEM results from line 154 to 160, the data are well displayed the morphologic changes and the differences in their killing kinetics after treatment with peptides of G6, A6 and PMB, However, two questions that may need to be answered:

(1) As did in the section of bactericidal kinetics experiment (line 124-125), it shows that similar to polymyxin B (PMB), $1 \times$ or $2 \times$ MIC A6 and G6 could completely kill *E. coli* CVCC195 within 1 h, more rapidly than N6 (4 h) (Fig. 2a), while in TEM experiment, after treatment for 2h, approximately 75% and 50% abnormal cells were observed in A6 and G6, but only 1% in PMB-treated groups, the data seem contradictory, please explain.

(2) In this work, it seems is that the major interest is the selectivity toward gram-negative bacteria after construction of chimeric peptides by adding an LPS-binding domain. If so, why did author(s) also add a gram-positive bacterium as a control in TEM or SEM experiments?

8. In SEM and TEM results, line 152-153 and line 159-160, the authors mentioned that the modes of action of A6 and G6 may be different from that of N6 and PMB, please explain in discussion or cite a literature.

9. The overall interest of this manuscript is on anti-endotoxic and anti-inflammatory activities of SCPs. These peptides were constructed by conjugating a targeting domain with a killing domain. However, through the whole manuscript, we didn't find any experimental data, explanation, or structural analyses, on the detailed function and role of each individual domain (LPS binding and Killing domains) within the sequence.

10. As indicated in the section of "Discussion" from line 335-346, regarding the enhanced binding affinity of A6 and G5 to LPS, why did authors not simply add a scramble peptide with positive charge(s) to the N6 domain as a control? It would be much easier to confirm the deduction and make thing clearer.

11. Line 351-352, the phrase "This prominent enhanced antibacterial and antiendotoxic activity of A6 and G6 was likely attributed to the effectiveness of flexible and rigid linkers", The structure of A6 and G6 are "LBP14-linker-N6", why did authors think "was likely attributed to the effectiveness of flexible and rigid linkers"? It seems lack of sufficient evidence. Please explain or revise it.

Minors:

Some language editing should be required, for example:

Line 61 "acitivies" should be "activities"

Line 76 "efficiently": should be efficient

Line 381 "domonstrated" should be demonstrated.

Response to the comment of Reviewer 1#

Question 1: Writing: The overall outline and organization of the paper is good but the text needs to be carefully edited for spelling, grammar and scientific language.

Answer 1: English description of this full text was polished in style of American English via Elsevier Language Editing Co. with a Project nr: 169208 at payment of 256.5 USD (its invoice No.: LE 254975), it was finished and sent back on Sep. 25, 2019.

Question 2: Reproducibility and transparency: The authors have performed detailed in vitro and in vivo experiments to support their claims. The level of detail in the methods section is adequate for repeating the experiments. However, the raw data is not available for the graphs. In addition, for the SEM and TEM figures, it is not clear if the selected images are representative for the whole community of cells. It would be useful to include the full scale images. It would also be useful to comment on the statistical significance of the mice experiments.

Answer 2: i) According to your requirements, all raw data of the graphs (Figs. 2b-2f and 2h; Figs. 4a-fc; Fig. 5a; Fig. 6b; Figs. 7b-7d; Supplementary Figs. 13a and 13b) were re-analyzed and the related statistical analysis was performed in the stability (temperatures, pH values, enzymes and serum), hemolysis, dissociation of LPS aggregates, BC probe displacement, LPS-LBP binding, cytokines in RAW 264.7 cells and all mice experiments, respectively.

Line 634-637: “**Statistical analysis.** All data are presented as means \pm standard deviation (SD). Statistical analyses between treatments or groups were determined using one-way analysis of variance (ANOVA) models in SAS 9.2 (SAS Institute Inc., Cary, NC, USA), followed by Dunnett's multiple comparisons test. A p -value of < 0.05 was considered statistically significant.”.

Line 829-830: “...cytotoxicity of A6, G6 and N6 against RAW 264.7 monocytes. Different lower case letters indicate a significant difference between two groups ($p < 0.05$).”.

Line 844: “...the peaks. Different lower case letters indicate a significant difference between two groups ($p < 0.05$).”.

Line 845-851: “**Fig. 5** Effects of SCPs-A6 and G6 on LPS binding to LBP and macrophages. **a** Effects of A6 and G6 on the binding of LPS and LBP. The 96-well microtiter plates were coated with 4 ng/ml LPS overnight at 4°C, washed three times with PBST, and blocked with PBS for 2 h. After

washing again, the plates were incubated with different concentrations of A6 and G6 at 37°C for 1 h. After the addition of primary and secondary antibodies, the absorbance was measured at OD_{450 nm}. N6 and PMB were used as controls. Different lower case letters indicate a significant difference between two groups ($p < 0.05$). **b** Location...”.

Line 864-866: “The mouse survival was recorded for 7 d. **b** Effects of A6 and G6 on the cytokines and IAP levels. The results are given as the mean \pm SD. Different lower case letters indicate a significant difference between two groups ($p < 0.05$).”.

Line 871-872: “The protein levels of p65, p-p65, ERK1/2, p-ERK1/2 and I κ B α in lungs were analyzed by Western blotting. Different lower case letters indicate a significant difference between two groups ($p < 0.05$).”.

Supplementary information:

Line 123-124: “... levels of TNF- α , IL-6 and IL-10 in supernatants were measured using ELISA. The data are given as the mean \pm SD. Different lower case letters indicate a significant difference between two groups ($p < 0.05$).”.

ii) In EM figures, the selected images are representative for the whole community of cells. The TEM with different scales were added into Fig. 3 in this revision to clearly display the whole community of cells. The revised Fig. 3 and other SEM with different scales (not shown in Fig. 3 in this revision) were shown in the followings:

Fig. 3

Other SEM images (not shown in Fig. 3 in this revision)

Response to the comment of Reviewer 2#

Question 1: In line 61, authors mentioned that “the activities of a few AMPs can be specifically improved by attaching a targeting region to generate novel specifically-targeted chimeric peptides that containing functionally independent targeting and killing moieties or domains”, while based on the data in Table 2, peptides A6 and G6 seem no strong gram-positive and negative selectivity, even worse than N6, a killing peptide itself, please explain.

Answer 1: i) You are right. In this study, A6 and G6 did not show improvement in MICs after 24 h incubation (Table 2), which is similar to the previous study that anti-*Pseudomonas* STAMP (G10KHc) (Echert et al. 2006). The MIC assay only measures the concentration of each peptide that is required to maximally inhibit bacterial culture growth, thus differences in the killing rate between SCPs-A6/G6 and their parent-N6 may be obscured. We anticipated that the targeting ability of LBP14 would help increase the killing rate of SCPs, relative to that of N6 alone, and that this effect would not extend to other species (Eckert et al. 2006).

ii) The selectivity experiment of A6, G6 and N6 was added into and the “Supplementary Information” and “Results” sections in this revision. The result showed that after treatment with A6 (0.77–0.59) or G6 (0.51–0) for 1–5 min, the ratio of the recovered *E. coli* to *S. aureus* cells was significantly lower than that of N6 (0.91–0.67) (Supplementary Fig. 4), which indicating that A6 and G6 could preferentially kill targeted bacteria (*E. coli*) in mixed bacterial cultures within a short time. This is consistent with the previous study that G10KHc selectively killed *Pseudomonas* in mixed bacterial cultures (Eckert et al. 2006).

iii) Except for the MIC values and killing selectivity, both A6 and G6 also displayed more rapidly kill *E. coli* (Fig. 2a), lower hemolysis (Fig. 2h), greater efficacy of dissociation of LPS aggregates (Fig. 4a), more potent binding to LPS/lipid A (Supplementary Table 3), more potent blocking the binding of LPS to LBP or to RAW264.7 cells (Fig. 5a and c), and higher survivors of mice challenged with LPS and *E. coli* (Fig. 7a) than N6. Therefore, A6 and G6 had higher antibacterial and antiendotoxic activity than N6 *in vitro* and *in vivo*, although they did not show improvement in MIC.

Reference:

Eckert, R., et al. Adding selectivity to antimicrobial peptides: rational design of a multidomain peptide against *Pseudomonas* spp. *Antimicrob Agents Chemother.* **50**(4), 1480–1488 (2006).

Supplementary Information Figure S4:

Supplementary Figure 4. Killing selectivity of peptides. A 1:1 mixture of *E. coli* (Ec) and *S. aureus* (Sa) was treated with A6 (16 $\mu\text{g/ml}$), G6 (16 $\mu\text{g/ml}$) or N6 (2 $\mu\text{g/ml}$) for 1, 3 and 5 min, respectively; survivors were counted in MH plates. The change in relative ratio of *E. coli* to *S. aureus* (Ec/Sa) after treatment with A6, G6 or N6 is shown. It was repeated for three times.

Line 153-160: “The mixed bacterial species were used to determine whether SCPs-A6 and G6 could exhibit selectivity for *E. coli*. Both A6 and G6 selectively killed *E. coli* in the mixed *E. coli*-*S. aureus* cultures. G6 displayed higher activity than A6 (Supplementary Fig. 4). After treatment for 1 or 5 min with A6 (ratio of 0.77–0.59) or G6 (ratio of 0.51–0), the ratio of *E. coli* to *S. aureus* cells was significantly lower than that of the N6-treated group (ratio of 0.91–0.67). This indicates that both A6 and G6 are able to preferentially kill the targeted bacteria in the mixed species cultures due to the attachment of the LPS-targeting domain LBP14, which is sufficient to guide the chimeric peptides to selectively bind to LPS on Gram-negative cells¹⁶.”.

Supplementary Information:

Line 22-34: “Mixed species killing kinetics of SCPs-A6 and G6

The mid-exponential phase *E. coli* CVCC195 and *S. aureus* ATCC43300 cultures were diluted to 2×10^5 CFU/ml, mixed (1:1) and added into the 96-well plates. A6, G6 or N6 were added into each well of plates to final concentration of 16 (A6 and G6) or 2 (N6) $\mu\text{g/ml}$ in a 200- μl total volume. An aliquot was removed at 1, 3 and 5 min, respectively, and survivors were counted on MH solid plates¹⁻³.

References:

1. Kaplan, C. W., et al. Selective membrane disruption: mode of action of C16G2, a specifically targeted antimicrobial peptide. *Antimicrob Agents Chemother.* **55**(7), 3446–3452 (2011).
2. Eckert, R., et al. Adding selectivity to antimicrobial peptides: rational design of a multidomain peptide against *Pseudomonas* spp. *Antimicrob Agents Chemother.* **50**(4), 1480–1488 (2006).
3. Eckert, R., et al. Targeted killing of *Streptococcus mutans* by a pheromone-guided "smart" antimicrobial peptide. *Antimicrob Agents Chemother.* **50**(11), 3651–3657 (2006).”.

Line 69-72: “**Supplementary Figure 4. Killing selectivity of peptides.** A 1:1 mixture of *E. coli* (Ec) and *S. aureus* (Sa) was treated with 16 µg/ml A6, G6 or N6 for 1, 3 and 5 min, respectively; survivors were counted in MH plates. The change in relative ratio of *E. coli* to *S. aureus* (Ec/Sa) after treatment with A6, G6 or N6 is shown. It was repeated for three times.”.

Question 2: As compared with a similar work cited in lit. 16 [Kim, H., Jang, J. H., Kim, S. C. & Cho, J. H. Enhancement of the antimicrobial activity and selectivity of GNU7 against Gram-negative bacteria by fusion with LPS-targeting peptide. *Peptides*, 82, 60–66 (2016), what’s the significance or novelty of this study?

Answer 2: In this study, we firstly designed SCPs-A6 and G6 based on the targeting domain (LBP14) and killing domain (N6) by two different linkers (flexible/rigid), compared effects of linkers on dual-function (antibacterial and neutralizing-LPS activity), evaluated the dual-function efficacy of SCPs *in vitro* and in mice, and preliminarily explored possible mechanisms of SCPs in killing MDR *E. coli* and neutralizing LPS. The details of differences between Ref. 16 and our work are shown in the followings:

i) In Reference 16, author compared different LPS-binding regions of LPS-binding proteins or peptides and designed three chimeric peptides including Syn-GNU7 with a flexible linker (GGG) based on three different LPS-targeting peptides to select proper LPS-targeting domains. The cytotoxicity, LPS-binding activity and selective antibacterial activity were detected *in vitro*, as well as effect of the hybrid peptides on LPS induced pro-inflammatory cytokine gene expression in RAW 264.7 cells.

The result showed that: ①among them, Syn-GNU7 displayed higher LPS-binding and lower MIC values (8–32-fold) than the parent peptide *in vitro* against Gram-negative bacteria *Escherichia coli* and *Salmonella typhimurium*; ②Syn-GNU7 could selectively eliminate Gram-negative bacteria from mixed culture. It indicated that LPS-targeting peptides could increase in targeting specificity rather than a generally enhanced killing ability. ③Syn-GNU7 had higher toxicity (nearly 20% hemolysis; nearly 80% viability in the MTT assay) than parents even at 8 µM.

However, the bactericidal kinetic, stability (such as toward temperature, digestive enzymes, serum and pH), the development of bacterial resistance and mechanism of Syn-GNU7 remain unclear, as well as *in vivo* antibacterial activity and LPS neutralization ability. Reference 16 provides the basis for construction technology of chimeric peptides for selection of proper LPS-binding regions, which is its novelty.

ii) In our study, except for above experiments included in Reference 16, NMR structures, effects on the development of bacterial resistance, bactericidal kinetic, tissue distribution in mice, antibacterial activity and LPS neutralization/depolymerization ability of SCPs-A6 and G6 *in vitro* and *in vivo*, effects on LPS cascades and their possible mode of actions were also carried out.

The result showed that: ①A6 and G6 retained β -sheet structures, which is similar to parent; ②A6 and G6 had similar or lower toxicity (100%, 81.1%) than parent (88.2%) at 128 $\mu\text{g/ml}$ and lower hemolysis (1.2%, 0.026%) than parent (1.9%) at 256 $\mu\text{g/ml}$; ③A6 and G6 remained the intrinsic antibacterial activity against MDR *E. coli* CVCC195 to high temperature, acidic and alkaline conditions, more rapidly killed *E. coli* (within 1 h) than parent (4 h); ④A6 and G6 had the stronger binding capacity to LBP14 and far more dissociation of LPS aggregates than parent; ⑤A6 and G6 displayed more potent antibacterial activity and anti-endotoxin *in vitro* and *in vivo*, respectively, which may be related to linkers that providing a spatially-appropriate arrangement for more efficient interactions between peptides and bacteria (Haga et al. 2013; Shamriz et al. 2016).

This work suggests that A6 and G6 may be promising dual-function candidates as novel antibacterial and anti-endotoxin agents to treat MDR *E. coli* and sepsis; it provides new clues for the design of dual-functional chimeric peptides by different linkers and gives the preliminary data support for preclinical/clinical studies, which is its significance.

References:

Haga, T., Hirakawa, H. & Nagamune, T. Fine tuning of spatial arrangement of enzymes in a PCNA-mediated multienzyme complex using a rigid poly-L-proline linker. *PLoS One* **8**, e75114 (2013).

Shamriz, S. & Ofoghi, H. Design, structure prediction and molecular dynamics simulation of a fusion construct containing malaria pre-erythrocytic vaccine candidate, PfCelTOS, and human interleukin 2 as adjuvant. *BMC Bioinformatics* **17**, 71 (2016).

Question 3: From line 73 to 76, authors mentioned the differences between “flexible linker” and “rigid linker”. However, through the whole manuscript, did author try a chimeric peptide that without a

“linker”? is it necessary in any cases to have a linker? though authors cited three literatures in discussion (line 319-321), however, there are also a lot of studies that show good data without a linker. Actually, it seems that no significant differences in all tests between A6 and G6 within this study. Please explain.

In addition, it seems authors put too much focus on “linkers” (line 68-79, line 89-90, line 318-319, line 319-331 etc.). If so, which one confers better results in this work? Is “flexible” one or “rigid” one? Any differences, such as secondary structure in aqueous solution, in the presence and absence of LPS, LPS-neutralization ability, anti-inflammatory activity, etc. between these two peptides? The discussion may necessarily be required.

Answer 3: i) According to your requirements, one chimeric peptide without linkers (LBPN6) was synthesized and its MICs were determined in this revision. The MICs of LBPN6 against *Escherichia* and *Salmonella* strains were 7.6–15.2 μM , respectively, significantly higher than that of A6 (1.3–3 μM) and G6 (0.88–3.52 μM), indicating that the linkers are necessary for activity of chimeric peptides in this study. The result was added into the Supplementary Table 1 and Table 2.

Supplementary Table 1 Amino acid sequences and physicochemical properties of LBP_{N6}, N6CK, A6CK and G6CK.

Name	Sequence	MW (Da)	PI	Charge (+)	GRAVY	AI	AAS	BI (kcal/mol)	AH (%) ^a
LBP _{N6}	LBP14-N6	4224.93	11.89	9	-0.503	58.29	0.333	2.87	20
N6CK	V C VYRGFAWN C HRRANNGVRV	2477.85	10.72	4	-0.310	64.76	0.229	2.61	23.81
A6CK	LBP14-(EA ₃ K) ₂ -N6CK	5165.99	11.33	9	-0.480	58.67	0.027	2.54	64.44
G6CK	LBP14-G ₄ S-N6CK	4540.22	11.89	9	-0.500	51.00	0.471	2.50	20

MW: molecular weight; PI: isoelectric point; GRAVY: grand average of hydropathicity; NN: no data; AI: aliphatic index; AAS: antibacterial activity score; BI: Boman index; AH: alpha helix; a: calculated by NPS@; underlined residues: disulphide bond. LBP14N6 without linkers was used as a control; N6CK, A6CK and G6CK were designed as scramble controls of N6, A6, and G6, respectively.

Table 2 MIC values of SCPs, PMB and control peptides.

Strains	MIC																	
	A6		G6		N6		PMB		LBP14		LBPN6		N6CK		A6CK		G6CK	
	µg/ml	µM	µg/ml	µM	µg/ml	µM	µg/ml	µM	µg/ml	µM	µg/ml	µM	µg/ml	µM	µg/ml	µM	µg/ml	µM
Gram-negative bacteria																		
Escherichia coli CVCC195 ^a	8	1.5	4	0.88	1	0.4	1	0.84	8	4.53	64	15.2	>64	>25.8	64	12.4	32	7.1
E. coli CVCC1515 ^a	16	3	16	3.52	1	0.4	1	0.84	8	4.53	32	7.6	>64	>25.8	>64	>12.4	16	3.5
Salmonella typhimurium ATCC14028 ^c	8	1.5	4	0.88	4	1.6	2	1.68	16	9.06	32	7.6	>64	>25.8	>64	>12.4	32	7.1
S. typhimurium CVCC533 ^a	16	3	16	3.52	2	0.8	1	0.84	32	18.1	32	7.6	>64	>25.8	>64	>12.4	>64	>14.2
S. pullorum CVCC1809 ^a	8	1.5	8	1.76	2	0.8	1	0.84	16	9.06	32	7.6	>64	>25.8	>64	>12.4	32	7.1
Pseudomonas aeruginosa CICC10419 ^b	>64	>12	16	3.52	16	6.4	>64	>53.76	64	36.2	64	15.2	>64	>25.8	>64	>12.4	>64	>14.2
Gram-positive bacteria																		
Staphylococcus aureus ATCC43300 ^c	>64	>12	16	3.52	16	6.4	>64	>53.76	64	36.2	64	15.2	>64	>25.8	>64	>12.4	>64	>14.2
Enterococcus faecalis CVCC3936 ^a	32	6	16	3.52	32	12.8	>64	>53.76	32	18.1	8	1.9	>64	>25.8	>64	>12.4	16	3.5
Streptococcus suis CVCC606 ^a	32	6	16	3.52	64	25.6	>64	>53.76	>64	>36.2	32	7.6	>64	>25.8	>64	>12.4	>64	>14.2
Bacillus subtilis ATCC6633 ^c	8	1.5	8	1.76	1	0.4	0.5	0.42	32	18.1	16	3.8	>64	>25.8	16	3.1	4	0.9
Listeria ivanovii ATCC19119 ^c	>64	>12	>64	>14.08	>64	>25.6	>64	>53.76	8	4.53	8	1.9	>64	>25.8	>64	>12.4	8	1.8
Fungi																		
Candida albicans CGMCC2.2411 ^d	>64	>12	>64	>14.08	>64	>25.6	>64	>53.76	>64	>36.2	>64	>15.2	>64	>25.8	>64	>12.4	>64	>14.2

Line 109-118: “LBP14N6 without linkers was used as a control; N6CK, A6CK and G6CK were designed as scramble controls of N6, A6, and G6, respectively. As shown in Table 1, compared with parents N6 (+4) and LBP14 (+5), LBP6, A6 and G6 (+9) have more positive charges, indicating a possible stronger electrostatic attraction with LPS than N6²⁶. The hydrophobicity and aliphatic index of LBP6, A6 and G6 were lower than that of N6. The Boman index of A6 and G6 decreased from 2.61 to 2.54 and 2.50 kcal/mol, respectively, and the α -helix content of LBP6, A6 and G6 was higher than N6, suggesting more potent antibacterial activity than N6²⁷. A larger positively charged cloud was observed in LBP6, A6 and G6 than in N6 (Supplementary Fig. 1a), indicating that they may interact more potently with bacterial membranes²⁸.”

Line 145-146: “LBP6 and G6CK showed very weak activity against the tested bacterial strains (MICs of 1.9–15.2 μ M); both N6CK and A6CK showed hardly any activity against all of the tested strains (Table 2).”.

Line 691-693: “28. Inácio, Â. S. et al. Quaternary ammonium surfactant structure determines selective toxicity towards bacteria: mechanisms of action and clinical implications in antibacterial prophylaxis. *J. Antimicrob. Chemother.* **71**, 641–654 (2016).”.

ii) Yes, you are right. There is no significant differences in antibacterial and antiendotoxic activity between A6 and G6 in this study. As a huge grey system *in vivo* and *in vitro* during biological and biochemical processes with various AMPs, multiple factors are involved in complex across action among them, we could logically expect and analyze that even though the most rigid experiment design by regular standard of experiment could not be revealed a panorama of action mechanism as whole in one time work, a reasonable deduction for impairment / incomplete or partial initial finding or observation should be buffered, don't you think so. On the one hand, the properties of A6 is superior to those of G6 in time-killing dynamics, displacing BC probe, the capacity to bind to LPS or lipid A, blocking LPS binding to LBP and regulating signal pathway in mice. The results showed that A6 had a more rapid bactericidal efficiency against *E. coli*, markedly less cytotoxicity, more potent capacity to bind to LPS/lipid A, block LPS binding to LBP and regulate signal pathway in mice than G6. On the other hand, however, the properties of G6 is superior to those of A6 in selectivity killing, hemolysis, serum stability, depolymerization of LPS polymers, inhibition of LPS binding to RAW 264.7 cells and survivor of mice challenged with *E. coli* or LPS. These differences were added into the “Results” and “Discussion” sections. Generally, the two linkers in chimeric peptides play an important role in remaining independent biological activities of targeting and killing domains of SCPs, but it is difficult

for us to give an accurate conclusion that which one is better than another, which may need more experiments to reveal and explain the discrepancy between A6 and G6 or different linkers in details in our next study in the further.

Line 155: “G6 displayed higher activity than A6 (Supplementary Fig. 4).”.

Line 176-179: “The cell survival of RAW 264.7 cells when exposed to A6, G6 and N6 was 100%, 81.1% and 88.2%, respectively, at a concentration of 128 µg/ml (Fig. 2i), indicating that A6 had markedly less cytotoxicity than G6 and N6.”

Line 210-215: “As shown in Fig. 4a, both A6 and G6 induced a larger fluorescence change than N6 and PMB in a dose-dependent manner, and G6 dissociated LPS aggregates with a greater efficacy than A6. These results indicate that the interaction of the SCPs with LPS may result in far more dissociation of the LPS aggregates than interaction with N6 or PMB, making it unavailable for LBP binding^{5,35}.”.

Line 310: “A6 more potently blocked the production of cytokines than G6 or N6.”

Line 327-328: “The survival rates of the mice treated with 4 µmol/kg A6 or G6 were 80% and 100%, identical or superior to N6 (80%).”.

Line 331-332: “For LPS injection, after treatment with 0.125 µmol/kg A6 or G6, the survival of the mice was 100% for both, which was higher than that of N6 (60%) (Fig. 6a).”.

Line 351-353: “After treatment with peptides, the expression of IAP in mice was markedly promoted by 15.5% (A6) and 11.5% (G6). This was higher than that promoted by N6 (4.5%), but lower than that by PMB (27.1%).”.

Line 400-402: “This indicated that the flexible and rigid linkers could maintain a certain distance between the domains, which contributed to the retention of their independent structures and biological activities¹⁷.”.

Question 4: Line 110-111, the structures of SCPs analyzed by NMR spectroscopic technology are reasonable. However, it should be pointed out that the data were acquired in aqueous solution, while the presence of organic solvent or lipid sometimes will induce a significant change in their secondary

structure. Therefore, authors have ever considered the analyses of their structures in the presence of lipid? It might be closer to the “real” situation when interacted with bacterial cell membrane.

Answer 4: i) Thanks a lot for the suggestion. “in aqueous solutions” was added into the Results “Physicochemical properties and structures of SCPs-A6 and G6.” section and Figure S1 in this revision.

Line 129-132: “The nuclear Overhauser effect spectroscopy (NOESY) spectra and DANGLE analysis of A6 and G6 indicated that the killing domain retains β -sheet structures, and the LPS-targeting LBP14 domain displays a random coil conformation without structural convergence in aqueous solutions (Supplementary Figs. 2 and 3).”.

Supplementary information

Line 54-57: “**Supplementary Figure 1. Structural analysis of peptides. a** Electrostatic potential surface of SCPs-A6, G6 and N6. Blue, red and white represent positive, negative and neutral charge, respectively. Molecular models were generated with PyMOL 1.8. **b** NMR analysis of A6, G6 and N6 in aqueous solutions.”.

ii) We actually considered to solve the SCPs’ structures in the presence of LPS and/or Lipid A. We collected the tr-NOE spectra to solve the structures with this method (Fig. 4d and e), but the tr-NOE spectra turned out to be very similar to the free form NOE with only a few additional cross peaks. Although NOE enhancement was observed in these tr-NOE spectra, the structure determination trails with the presence of LPS gave very similar 3D structures to the free forms. Additionally, tr-NOE experiments in the presence of lipid A showed little enhancement (see **Figures a-e in the following**; note that lipid A has stronger interaction to A6/G6 when comparing to LPS). Considering the dissociation constants when binding to LPS/lipid A were in low μ M level, the exchange could be too slow to get good tr-NOE spectra and the tr-NOE cross peaks could be seriously contaminated by free form NOEs. We tried to increase the experiment temperature to 40 degree Celsius (**Figures b and e in the following**) and use our 500 MHz instrument (**Figure e in the following**) to fulfill the tr-NOE conditions, but neither worked. So we finally gave up the structure determination with the presence of lipid A and discussed the structure changes only using the CD data.

Figure a. Amide region of NOESY spectra of the A6 sample in the absence (black) and presence (red) of lipid A. The molar ratio of peptide to lipid A is ca. 40 to 1. Spectra were collected at 298 K on a 600 MHz instrument with a mixing time of 100 ms.

Figure b. Amide region of NOESY spectra of the A6 sample in the absence (black) and presence (red) of lipid A. The molar ratio of peptide to lipid A is ca. 40 to 1. Spectra were collected at 313 K on a 600 MHz instrument with a mixing time of 100 ms.

Figure c. Amide region of NOESY spectra of the G6 sample in the absence (black) and presence (red) of lipid A. The molar ratio of peptide to lipid A is ca. 40 to 1. Spectra were collected at 298 K on a 600 MHz instrument with a mixing time of 100 ms.

Figure d. Amide region of NOESY spectra of the G6 sample in the absence (black) and presence (red) of lipid A. The molar ratio of peptide to lipid A is ca. 20 to 1. Spectra were collected at 298 K on a 600 MHz instrument with a mixing time of 100 ms.

Figure e. Amide region of NOESY spectra of the G6 sample in the absence (black) and presence (red) of lipid A. The molar ratio of peptide to lipid A is ca. 20 to 1. Spectra were collected at 313 K on a 500 MHz instrument with a mixing time of 100 ms.

Question 5: Line 115-116, the phrase “which may be due to the absence of amphipathic conformations of the peptides when interacted with bacteria”, please provide any possible reason in discussion. Why?

Answer 5: The wrongly descriptive sentence “which may be due to the absence of ... when interacted with bacteria” was deleted from the text in this revision.

Line 135-137: “The MICs of A6 and G6 (0.88–3.52 μM) did not show improvement when compared to N6 (0.4–1.6 μM) against most Gram-negative bacteria, such as *E. coli* and *Salmonella typhimurium*, ...”.

Question 6: Line 125-130, the phrase “The result indicated that the antibacterial activity of A6 and G6 is consistent with a bactericidal mode of action.” Please explain, or provide exact literature.

Answer 6: We did not properly describe the phrase “The result indicated that the antibacterial activity of A6 and G6 is consistent with a bactericidal mode of action”, so it was changed to “The result indicated that A6, G6 and PMB display a rapid bactericidal effect against *E. coli*”.

Line 151-152: “These results indicate that A6, G6 and PMB display a rapid bactericidal effect against *E. coli*”.

Question 7: In the section of TEM results from line 154 to 160, the data are well displayed the morphologic changes and the differences in their killing kinetics after treatment with peptides of G6, A6 and PMB, However, two questions that may need to be answered:

(1) As did in the section of bactericidal kinetics experiment (line 124-125), it shows that similar to polymyxin B (PMB), $1 \times$ or $2 \times$ MIC A6 and G6 could completely kill *E. coli* CVCC195 within 1 h, more rapidly than N6 (4 h) (Fig. 2a), while in TEM experiment, after treatment for 2 h, approximately 75% and 50% abnormal cells were observed in A6 and G6, but only 1% in PMB-treated groups, the data seem contradictory, please explain.

(2) In this work, it seems is that the major interest is the selectivity toward gram-negative bacteria after construction of chimeric peptides by adding an LPS-binding domain. If so, why did author(s) also add a gram-positive bacterium as a control in TEM or SEM experiments?

Answer 7: i) Thanks for your suggestion. The description of EM was rewritten in this revision. The abnormal cells in EM experiments refer to ones with lysis, uniform cytoplasmic electron and morphologically abnormal cells, in which at least 200 cells were visualized and scored for the observed characteristics according to the previous method (Otto et al. 2010). These abnormal cells may be only one part of dead cells, not all dead cells; the normal cells observed in EM with less obvious characteristics may be dead ones. 75%, 50% and 1% appeared in TEM only refer to the abnormal cell ratio (one part of dead cells), not the ratio of all dead cells. The SEM and TEM result displayed the cellular morphologic changes and the possible discrepancy in mode of action between A6/G6 and PMB. Comparably, however, the survivors of *E. coli* and killing rates were displayed in the bactericidal kinetics experiment; the result showed that similar to PMB, A6 and G6 had a rapid killing activity (completely kill *E. coli* within 1 h). So, the data are not contradictory.

Reference:

Otto, C. C., et al. Effects of antibacterial mineral leachates on the cellular ultrastructure, morphology,

and membrane integrity of *Escherichia coli* and methicillin-resistant *Staphylococcus aureus*. *Ann Clin Microbiol Antimicrob.* **9**, 26 (2010).

Line 184-190: “However, after treatment with A6 and G6, filamentous and extracellular polymeric substances appeared outside the cells, along with the leakage of contents, indicating a loss of membrane integrity, with the subsequent release of intracellular constituents³⁰. Noticeably, some collapsed lamellar cells were seen upon N6 treatment. In the PMB-treated group, numerous protrusions or blebs and filiferous substances appeared outside of the cells (Fig. 3a), which was similar to previous reports^{31,32}. ”.

Line 199-201: “Approximately 75% and 50% abnormal cells (including lysis, non-uniform cytoplasmic electron and morphologically abnormal cells) were observed with A6 and G6 treatment, which were higher than with N6 treatment (19%), indicating their ...”.

Line 547-548: “The samples were observed in a JEM1400 (JEDL, Tokyo, Japan)⁷³. At least 200 bacterial cells were visualized and scored for the observed characteristics⁷⁴. ”.

Line 791-793: “74. Otto, C. C., Cunningham, T. M., Hansen, M. R. & Haydel, S. E. Effects of antibacterial mineral leachates on the cellular ultrastructure, morphology, and membrane integrity of *Escherichia coli* and methicillin-resistant *Staphylococcus aureus*. *Ann Clin Microbiol Antimicrob.* **9**, 26 (2010).”.

ii) The Gram-negative selectivity killing experiment of SCPs was added into this revision, as shown in Answer 1. The result showed that A6 and G6 could preferentially kill targeted bacteria (*E. coli*) in mixed bacterial cultures within a short time.

iii) The experiment of *S. aureus* ATCC43300 used as a control in SEM was performed and added into “Results” and “Materials and methods” sections in this revision. Supplementary Fig. 5 was added into the Supplementary information, as shown in the following:

Line 190-194: “For MDR *S. aureus* CVCC43300, after treatment with A6, G6 or N6, many blebs, content leakage and lamellar collapses were observed outside of the *S. aureus* cells (Supplementary Fig. 5). In the untreated and PMB-treated controls, normal cell morphology and intact cell surfaces were apparent. This implied that the mode of action of A6 and G6 against MDR *E. coli* and *S. aureus* is

different from that of N6 or PMB³².”

Line 541-542: “SEM. Peptides (A6, G6 and N6) or PMB (4 × MIC) were added into mid-log phage MDR *E. coli* CVCC 195 and *S. aureus* CVCC43300 cells and were incubated at 37°C for 2 h.”.

Supplementary Figure 5. Effects of SCPs-A6 and G6 on the cell morphology and ultrastructure of MDR *S. aureus* CVCC43300. Bacteria in mid-logarithmic growth were treated with peptides or PMB at 4 × MIC for 2 h. Red arrows indicated typical disruptions (blebs, leakage of contents, and sheets), which were caused by peptides or PMB.

Question 8: In SEM and TEM results, line 152-153 and line 159-160, the authors mentioned that the modes of action of A6 and G6 may be different from that of N6 and PMB, please explain in discussion or cite a literature.

Answer 8: The description of SEM and TEM results was rewritten and the references were added into the “SCPs-A6 and G6 induced morphologic changes in *E. coli*” section in this revision.

Line 184-190: “However, after treatment with A6 and G6, filamentous and extracellular polymeric substances appeared outside the cells, along with the leakage of contents, indicating a loss of membrane integrity, with the subsequent release of intracellular constituents³⁰. Noticeably, some collapsed lamellar cells were seen upon N6 treatment. In the PMB-treated group, numerous protrusions or blebs and filiferous substances appeared outside of the cells (Fig. 3a), which was similar to previous reports^{31,32}. ”.

Line 197-206: “After treatment with 4 × MIC of A6, G6 or N6 for 2 h, heterogeneous electron density, disappearance of the outer and inner cell membranes, leakage of cellular contents and ghosts were observed in the *E. coli*. Approximately 75% and 50% abnormal cells (including lysis, non-uniform cytoplasmic electron and morphologically abnormal cells) were observed with A6 and G6 treatment, which were higher than with N6 treatment (19%), indicating their interaction with bacterial membranes, the subsequent leakage of cell contents and lysis³³. However, the least amount of cell lysis and ghosts was seen in response to PMB treatment (1%), which was in good accordance with the SEM images. The results indicated that there may be differences in how A6, G6, N6 and PMB interact with the *E. coli* cell membrane^{32,33}. ”.

Line 697-705: “30. Zajmi, A., et al. Ultrastructural study on the antibacterial activity of artonin E versus streptomycin against *Staphylococcus aureus* strains. *PLoS One* **10**, e0128157 (2015).

31. Dixon, R. A. & Chopra, I. Leakage of periplasmic proteins from *Escherichia coli* mediated by polymyxin B nonapeptide. *Antimicrob Agents Chemother.* **29**, 781–788 (1986).

32. Farkas, A., Maróti, G., Kereszt, A. & Kondorosi, É. Comparative analysis of the bacterial membrane disruption effect of two natural plant antimicrobial peptides. *Front Microbiol.* **8**, 51 (2017).

33. Díaz-Visurraga, J., Cárdenas, G. & García, A. “Morphological changes induced in bacteria as evaluated by electron microscopy”, in *Microscopy: Science, Technology, Applications and Education*, Méndez-Vilas, A., Díaz Álvarez, J., Eds. (Formatex Research Center, 2010), vol. 3, pp. 307–315.”.

Question 9: The overall interest of this manuscript is on anti-endotoxic and anti-inflammatory activities of SCPs. These peptides were constructed by conjugating a targeting domain with a killing domain. However, through the whole manuscript, we didn't find any experimental data, explanation, or structural analyses, on the detailed function and role of each individual domain (LPS binding and Killing domains) within the sequence.

Answer 9: According to your requirements, the roles, experimental data, explanation of each individual domain (LPS binding and Killing domains) of SCPs in antibacterial activities, killing selectivity, binding to LPS, LPS binding to LBP, depolymerization of LPS polymers, the affinity of peptides to macrophages and efficacy in mice were added into the "Results" and "Discussion" section.

Line 145-148: "LBPN6 and G6CK showed very weak activity against the tested bacterial strains (MICs of 1.9–15.2 μ M); both N6CK and A6CK showed hardly any activity against all of the tested strains (Table 2). These findings suggest that the antibacterial activity of chimeric peptides is associated with the killing domain (N6) and the linkers, which is similar to previous studies^{16,17,20}."

Line 153-160: "The mixed bacterial species were used to determine whether SCPs-A6 and G6 could exhibit selectivity for *E. coli*. Both A6 and G6 selectively killed *E. coli* in the mixed *E. coli*-*S. aureus* cultures. G6 displayed higher activity than A6 (Supplementary Fig. 4). After treatment for 1 or 5 min with A6 (ratio of 0.77–0.59) or G6 (ratio of 0.51–0), the ratio of *E. coli* to *S. aureus* cells was significantly lower than that of the N6-treated group (ratio of 0.91–0.67). This indicates that both A6 and G6 are able to preferentially kill the targeted bacteria in the mixed species cultures due to the attachment of the LPS-targeting domain LBP14, which is sufficient to guide the chimeric peptides to selectively bind to LPS on Gram-negative cells¹⁶."

Line 252-254: "These findings indicated that both A6 and G6 have a stronger binding capacity to LPS than N6 or LBP14, which may be ascribed to the significant binding of LBP14 to LPS²⁴."

Line 275-279: The LPS-LBP binding rates were reduced to 68–17% (A6) and 79–13% (G6), lower than those of N6 (82–56%) and PMB (82–22%), indicating a more potent ability of A6 and G6 to inhibit LPS-LBP interaction compared to N6 and PMB. This may be associated with an appropriate distance between the functional domains provided by the linkers, thus retaining their biological activity^{39,40}."

Line 291-293: “Additionally, the affinity of A6 and G6 to macrophages was higher than that of N6 (Fig. 5b), which may be ascribed to the ability of LBP14 to bind to LPS²⁴.”.

Line 310-313: “This suggests that both A6 and G6 significantly inhibit LPS-induced TNF- α , IL-6, and IL-10 production, which may be due to the potent ability of LBP14 to bind to LPS and further inactivate the related downstream MAPK cascade⁴².”.

Line 395-398: “Similarly, in our study, the higher MIC values of LBPN6 without a linker compared to the MICs of A6 and G6 indicated its very weak antibacterial activity, which may be related to a relatively limited spatial distance between the targeting and killing domains, thereby affecting their independent biofunctions (Table 2; Supplementary Fig. 1).”.

Line 351-355: “After treatment with peptides, the expression of IAP in mice was markedly promoted by 15.5% (A6) and 11.5% (G6). This was higher than that promoted by N6 (4.5%), but lower than that by PMB (27.1%). These results indicate that SCPs can better relieve the toxicity of LPS than N6 due to the addition of LBP14.”.

Line 400-406: “This indicated that the flexible and rigid linkers could maintain a certain distance between the domains, which contributed to the retention of their independent structures and biological activities¹⁷. The bactericidal kinetics and EM images showed that both A6 and G6 more rapidly and selectively killed *E. coli* and more seriously damaged bacterial cells than N6 alone, and even PMB (Figs. 2a and 3; Supplementary Fig. 4), which may be related to the increased interaction between the SCPs and the Gram-negative bacteria through binding to LPS¹⁶.”.

Line 433-435: “...leading to destabilization of the LPS assembly and a reduction in toxicity⁵⁴; this may be related to the ability of LBP14 to bind to LPS and depolymerize LPS polymers into monomers⁵⁵.”.

Question 10: As indicated in the section of “Discussion” from line 335-346, regarding the enhanced binding affinity of A6 and G5 to LPS, why did authors not simply add a scramble peptide with positive charge(s) to the N6 domain as a control? It would be much easier to confirm the deduction and make thing clearer.

Answer 10: i) Many thanks for your suggestion. Several scramble peptides of N6, A6 and G6 were designed as N6CK, A6CK and G6CK, respectively and synthesized by MIMOTOPES The Peptide

Company (Wuxi, China); the possible interaction between scrambles and LPS or lipid A was assessed by MD, CD, and SPR, respectively. The results showed that the enhanced binding affinity of A6 and G6 to LPS are not related to more positive charges, less random coils and more special residues or motifs such as Arg and Asn in SCPs. So, the description of deduction “This result may be ascribed to several factors: 1) more positive charges of A6... and 4) more special residues or motifs ...mainly by Arg10/Arg19/Asn21 residues^{25,45}.” was changed to “These results may be ascribed to the attachment of LBP14 to N6, which may enhance the LPS-binding ability²⁴.”.

ii) The methods and results of scramble peptides were added into the “Materials and methods”, “Results”, and “Supplementary information” (Supplementary Table 2 and 3; Supplementary Fig. 1, 6 and 8-10) sections, as shown in the following:

Line 410-414: “These results may be ascribed to the attachment of LBP14 to N6, which may enhance its LPS-binding ability²⁴. It has been demonstrated that Arg-Trp-Lys in LBP14, a BZB motif (B: basic amino acids, Z: arbitrary amino acids), is often observed in some LPS-binding molecules and proteins⁵⁰, suggesting that A6 and G6 are likely to have more LPS-binding sites than N6⁵¹.”.

Supplementary Table 2 CD analysis of secondary structures of SCPs and control peptides in the presence or absence of LPS.

Secondary structures	The percentages of secondary structures (%)															
	A6	A6+LPS	G6	G6+LPS	N6	N6+LPS	LBP14	LBP14+LPS	LBPN6	LBPN6+LPS	N6CK	N6CK+LPS	A6CK	A6CK+LPS	G6CK	G6CK+LPS
α -Helix	12.50	16.26	10.97	14.03	13.46	33.3	11.29	26.17	12.48	14.17	12.78	10.46	12.66	13.77	12.26	17.12
Antiparallel	35.45	23.22	35.85	24.39	46.94	1.73	40.25	12.03	24.72	20.03	42.85	45.20	33.48	30.82	40.45	27.30
Parallel	8.70	9.95	9.08	10.43	6.19	9.54	8.13	9.36	11.76	11.97	7.19	7.44	9.05	9.21	7.79	8.93
β -Turn	15.72	15.94	15.09	15.26	17.18	12.49	15.64	16.52	17.09	16.69	16.41	15.96	15.60	15.75	16.08	16.63
Random coil	27.63	34.63	29.00	35.97	16.23	42.93	24.68	35.91	34.02	37.13	20.70	20.86	29.21	30.44	23.43	30.02

Supplementary Table 3 Binding of SCPs and control peptides to LPS or lipid A in SPR.

Analytes	Ligands	kd (1/Ms)	kd (1/s)	KD (M)
LPS	A6	1.56×10^3	1.20×10^{-3}	7.68×10^{-7}
	G6	1.47×10^3	1.24×10^{-3}	8.46×10^{-7}
	N6	1.03×10^3	4.86×10^{-3}	4.71×10^{-6}
	PMB	6.66×10^3	3.02×10^{-4}	4.54×10^{-8}
	LBP14	1.91×10^1	1.32×10^{-3}	6.90×10^{-5}
	LBPN6	4.59×10^2	5.26×10^{-3}	1.15×10^{-5}
	N6CK	NN	NN	NN
	A6CK	1.87×10^2	4.84×10^{-4}	2.59×10^{-6}
	G6CK	1.42×10^1	1.43×10^{-3}	1.00×10^{-4}
Lipid A	A6	5.42×10^2	2.98×10^{-5}	5.49×10^{-8}
	G6	3.79×10^2	2.49×10^{-5}	6.57×10^{-8}
	N6	5.40×10^2	1.52×10^{-4}	2.81×10^{-7}
	G6CK	1.33×10^2	1.82×10^{-7}	1.39×10^{-9}

NN: no data; peptides could not bind with LPS or lipid A.

Supplementary Figure 1. Structural analysis of peptides. **a** Electrostatic potential surface of SCPs-A6, G6 and N6. Blue, red and white represent positive, negative and neutral charge, respectively. Molecular models were generated with PyMOL 1.8. **b** NMR analysis of A6, G6 and N6 in aqueous solutions. **c** Structures of scramble peptides analyzed by I-TASSER (Version 5.1) and Phyre (Version 2.0). LBP14N6 without linkers was used as a control; N6CK, A6CK and G6CK were designed as scramble controls of N6, A6, and G6, respectively.

Supplementary Figure 6. CD spectra for peptides with or without *E. coli* LPS (0.2 mg/ml). a LBP. b N6CK. c LBP6. d A6CK. e G6CK. f LPS.

Supplementary Figure 8. Molecular docking (MD) of peptides and LPS interaction. Docking was performed using Autodock4.2. Left: complex structures of LBP14 (**a**), LBPN6 (**b**), A6CK (**c**) or G6CK (**d**) and LPS. Oxygen, carbon, hydrogen atoms are indicated as red, green and white, respectively. Right: the residues participating in hydrogen bonding in peptides. The receptor (LPS) and peptide chains of A6 or G6 are shown as green and blue, respectively. The hydrogen bonds are indicated as red dotted lines.

Supplementary Figure 9. SPR analysis of the interaction between peptides and immobilized ligands-LPS. Sensorgrams indicated the association and dissociation phases of the interactions between LPS and peptides or antibiotic (LBP14, N6, LBPN6, A6, G6, N6CK, A6CK, G6CK and PMB) at different concentrations (8, 4, 2, 1, 0.5, 0.25, and 0.125 mM) from top to bottom. The sensorgrams were fit with a 1:1 binding kinetic model.

Supplementary Figure 10. SPR analysis of the interaction between peptides and immobilized ligands-lipid A. Sensorgrams indicated the association and dissociation phases of the interactions between lipid A and peptides or antibiotic (LBP14, N6, LBPN6, A6, G6, N6CK, A6CK, G6CK and PMB) at different concentrations (8, 4, 2, 1, 0.5, 0.25, and 0.125 mM) from top to bottom. The sensorgrams were fit with a 1:1 binding kinetic model.

Question 11: Line 351-352, the phrase “This prominent enhanced antibacterial and antiendotoxic activity of A6 and G6 was likely attributed to the effectiveness of flexible and rigid linkers”, The structure of A6 and G6 are “LBP14-linker-N6”, why did authors think “was likely attributed to the effectiveness of flexible and rigid linkers”? It seems lack of sufficient evidence. Please explain or revise it.

Answer 11: This improper description was changed to “This prominent enhanced antibacterial and antiendotoxic activity of A6 and G6 was likely attributed to the independent biological activities of targeting and killing domains separated by flexible and rigid linkers at distances and the correct folding of SCPs, as proven in NMR analysis (Supplementary Fig. 1b, 2 and 3)¹⁷.”.

Line 421-424: “This prominent enhanced antibacterial and anti-endotoxic activity of A6 and G6 can likely be attributed to the independent biological activities of the targeting and killing domains that are adequately separated by flexible and rigid linkers, along with the correct folding of the SCPs, as proven by NMR analysis (Supplementary Figs. 1b, 2 and 3)¹⁷.”.

Question 12: Minors:

Some language editing should be required, for example:

Line 61 “acitivies” should be” activities”

Line 76 “efficiently”): should be efficient

Line 381 “domonstrated” should be demonstrated.

Answer 12: i) English description of this full text was polished in style of American English via Elsevier Language Editing Co. with a Project nr: 169208 at payment of 256.5 USD (its invoice No.: LE 254975), it was finished and sent back on Sep. 25, 2019.

ii) “acitivies” was changed to “activities”.

iii) “efficiently” was changed to “efficient”.

iv) “domonstrated” was changed to “demonstrated”.

Line 63-65: “Therefore, the activities against the desired bacterium of some AMPs have been specifically improved by attaching a targeting region to generate novel, specifically targeted chimeric peptides with little impact on the normal flora;...”.

Line 83-84: “...retain a fixed distance between the functional domains, which may be more efficient than the flexible linkers^{21,22}.”.

Line 452-453: “It has been demonstrated that IAP has an anti-inflammatory affect due to its detoxification of LPS, ...”.

REVIEWERS' COMMENTS:

Reviewer #1 (Remarks to the Author):

The authors have significantly improved their manuscript by adding new experiments as well as new discussion of their results. However, two questions are still unclear in my mind:

1. The novelty of the findings. Some explanation of the significance is provided in the rebuttal letter but the novelty, if any, of the findings should also be outlined in the manuscript in comparison with previously reported chimeric peptides. This will also provide clues toward a mechanistic explanation for the action of these peptides.
2. The discussion related to the mechanism. This paragraph from the rebuttal letter is very confusing: "As a huge grey system in vivo and in vitro during biological and biochemical processes with various AMPs, multiple factors are involved in complex across action among them, we could logistically expect and analyze that even though the most rigid experiment design by regular standard of experiment could not be revealed a panorama of action mechanism as whole in one time work, a reasonable deduction for impairment / incomplete or partial initial finding or observation should be buffered, don't you think so."

Reviewer #2 (Remarks to the Author):

no more comments, all questions are answered.

Response to the comment of Reviewer 1#

Question 1: The novelty of the findings. Some explanation of the significance is provided in the rebuttal letter but the novelty, if any, of the findings should also be outlined in the manuscript in comparison with previously reported chimeric peptides. This will also provide clues toward a mechanistic explanation for the action of these peptides.

Answer 1: Many thanks for your suggestion. Some findings in our study were added the “Discussion” section to compare the details of differences between previous and our work. The novelty of this study was also indicated in the “Introduction” section.

Line 73-76: “However, these studies only provide a basis for the technology in which target-specific CPs were generated against some limited bacterial species, and little attention has been given to their toxicity, resistance, *in vivo* antibacterial/anti-endotoxic activity.”.

Line 421-431: “This prominent enhanced dual-function of A6 and G6 can likely be attributed to the linkers that providing a spatially-appropriate arrangement for more efficient interactions between peptides and bacteria (Supplementary Figs. 3b, 4 and 5). Notably, MDR *E. coli* CVCC195 did not develop resistance to SCPs-A6 or G6 after 30 passages, and they displayed low toxicity (Fig. 2g, h and i), which is superior to other CPs such as Syn-GNU7¹⁶.”

Line 476-479: “These results suggest that the SCPs may be promising dual-function candidates as novel antibacterial and anti-endotoxin agents to treat MDR *E. coli* and sepsis; it provides new clues for the design of dual-functional chimeric peptides by different linkers and gives the preliminary data support for preclinical/clinical studies.”.

Question 2: The discussion related to the mechanism. This paragraph from the rebuttal letter is very confusing: "As a huge grey system *in vivo* and *in vitro* during biological and biochemical processes with various AMPs, multiple factors are involved in complex across action among them, we could logistically expect and analyze that even though the most rigid experiment design by regular standard of experiment could not be revealed a panorama of action mechanism as whole in one time work, a reasonable deduction for impairment / incomplete or partial initial finding or observation should be buffered, don't you think so."

Answer 2: Very sorry for our improper explanation in the rebuttal letter. It means “There are many factors affecting the results in biological experiments; thus, for us, it is very difficult to provide all data

or exactly reveal a panorama of the action mechanism in details in this work, even if they are logistically designed to be perfect. We hope the partial data or deduction in this study may be useful, helpful for the design of novel, potent, dual-functional chimeric peptides.”.